# Are Large Reasoning Models Good Translation Evaluators? Analysis and Performance Boost

**Runzhe Zhan**[1]    **Zhihong Huang**[1]    **Xinyi Yang**[1]
**Lidia S. Chao**[1]    **Min Yang**[2]    **Derek F. Wong**[1✉]

[1]NLP[2]CT Lab, Department of Computer and Information Science, University of Macau
[2]Shenzhen Institute of Advanced Technology, Chinese Academy of Sciences
`nlp2ct.{runzhe,zhihong,xinyi}@gmail.com`
`min.yang@siat.ac.cn,{derekfw,lidiasc}@um.edu.mo`

## Abstract

Recent advancements in large reasoning models (LRMs) have introduced an intermediate "thinking" process prior to generating final answers, improving their reasoning capabilities on complex downstream tasks. However, the potential of LRMs as evaluators for machine translation (MT) quality remains underexplored. We provides the first systematic analysis of LRM-as-a-judge in MT evaluation. We identify key challenges, revealing LRMs require tailored evaluation materials, tend to "overthink" simpler instances and have issues with scoring mechanisms leading to overestimation. To address these, we propose to calibrate LRM thinking by training them on synthetic, human-like thinking trajectories. Our experiments on WMT24 Metrics benchmarks demonstrate that this approach largely reduces thinking budgets by $\sim$35x while concurrently improving evaluation performance across different LRM scales from 7B to 32B (e.g., R1-Distill-Qwen-7B achieves a +8.7 correlation point improvement). These findings highlight the potential of efficiently calibrated LRMs to advance fine-grained automatic MT evaluation.

Code: ⭘ ThinMQM        Models: 🤗 ThinMQM Models

## 1 Introduction

The development of reliable automated metrics that accurately mimic human judgments of translation quality is vital for advancing machine translation (MT) models [1, 2, 3, 4, 5]. However, modeling the comprehensive and complex "evaluation process" inherent in human assessment is challenging. Previous research has explored various paradigms to capture this process, from deterministic rule-based metrics [6, 7] and embedding-space similarities [8] to end-to-end neural networks [9, 10]. More recently, the rise of large language models (LLMs) as a judge [11, 12] has marked a substantial leap forward. LLMs provide a convenient mechanism for customizing the evaluation process through both natural language, marking a significant advancement in evaluation methodology [13, 14, 15].

Despite these advancements, assessing translation quality is rarely a simple "0-1" binary matching task. It often requires a deliberate, analytical cognitive effort, which is akin to "System 2" thinking [16], even for human annotators [17]. This suggests that emerging large reasoning models (LRMs) [18], which enhance reasoning capabilities by generating intermediate "thoughts" before producing final solutions, similar to human reflective thinking, may offer a stronger foundation for modeling the complex process of MT evaluation. While LRMs have often demonstrated remarkable performance boosts in solving mathematical and scientific challenges [19, 20], their potential as judges, i.e., LRM-as-a-judge in the specific context of MT evaluation remains largely unexplored.

---

✉Corresponding author.

39th Conference on Neural Information Processing Systems (NeurIPS 2025).

To provide a comprehensive understanding and practical insights into the application of LRM-as-a-judge for MT evaluation, this paper is the first to systematically address the following key questions: 1) *How do current LRMs perform in MT evaluation tasks when compared to human judgments?* 2) *What are the specific failure modes or inefficiencies encountered when applying LRMs to MT evaluation?* 3) *How can we develop an efficient and effective alignment strategy to tailor LRMs specifically for MT evaluation?*

To this end, we employ LRMs in MT evaluation under the multidimensional quality metrics (MQM) framework [21, 22], following previous state-of-the-art LLM-as-a-judge design principles [13, 14]. We then perform a meta-evaluation and analysis across a wide range of model series and sizes, including DeepSeek-R1 671B [19], QwQ 32B [23], and R1-Distilled models [19]. Through careful examination of critical factors in designing LRM-as-a-judge, our findings reveal several key insights. We observe a disagreement between LLMs and LRMs in their perception of evaluation materials, with strong LRMs benefiting more significantly from alignment with human-like evaluation protocols. Our results also suggest a need to rethink the design of multi-stage scoring mechanisms, as there are pitfalls related to overestimation problems and ambiguous contributions from auxiliary scoring models. Moreover, concerning thinking behaviors, we reveal that LRMs are not always efficient in allocating their thinking budget and tend towards "overthinking" for easier evaluation instances.

Furthermore, based on these findings, we propose a simple yet effective method to steer LRM perform **Thin**king-calibrated **MQM** (ThinMQM) scoring by training them on synthetic evaluation trajectories designed to mimic human-like scoring rubrics. Experimental results on the most recent WMT24 Metrics benchmarks [24, 25] show that this method can largely reduce thinking budgets by approximately ∼35x while improving the evaluation performance of LRMs at different model scales (notably, R1-Distill-Qwen-7B achieves a +8.7 correlation point improvement), as shown in Figure 1. Follow-up analysis reveals that such trajectory steering calibrates the scoring distribution and reduces the overestimation problem. These results align with our analysis, revealing substantial potential in developing LRM-as-a-judge for MT evaluation, yet highlight the necessity of controlling thinking budgets and performing careful calibration to fully realize this promise.

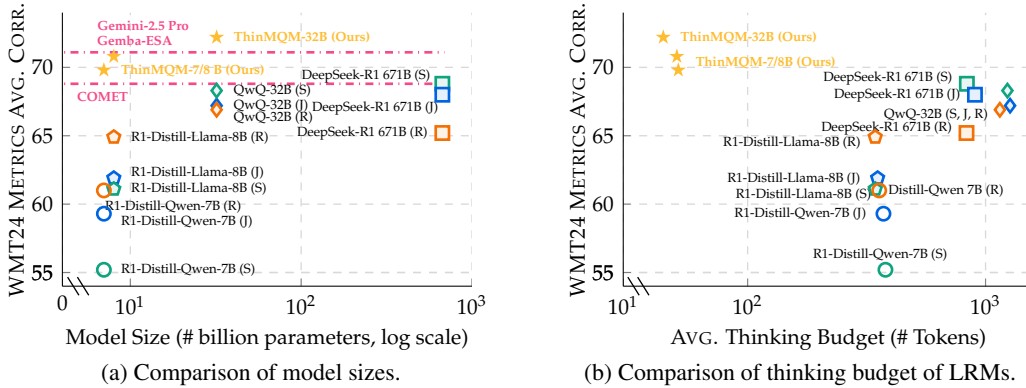

(a) Comparison of model sizes.

(b) Comparison of thinking budget of LRMs.

Figure 1: Performance comparison of various evaluation models on WMT24 metrics tasks. The ThinMQM models (Ours) achieve strong performance with competitive efficiency. "S, R, J" denote different evaluation inputs: Source-only, Reference-based, and Joint evaluation materials, respectively.

## 2 Preliminaries

### 2.1 Paradigms in Machine Translation Evaluation

Formally, traditional automatic evaluation metrics for machine translation can be abstractly defined as a mapping $m$ from a set of input materials $X$ to an output score $y$. The metric $m$ may take the form of a rule-based text matching algorithm [6, 7, 26, 27] or a parameterized model [8, 9, 28, 29, 10]. Typically, the input set $X$, i.e., the materials required for machine translation evaluation, comprises three elements: the machine translation hypothesis $h$, the reference translation $r$, and the source text $s$ in the original language. Evaluation is generally based on combinations of these elements, falling into two main categories: reference-based and reference-free. In the reference-based setting, the model hypothesis is compared directly with the reference translation (*Ref.*), i.e., $X_{\text{Ref.}} = (h, r)$. In

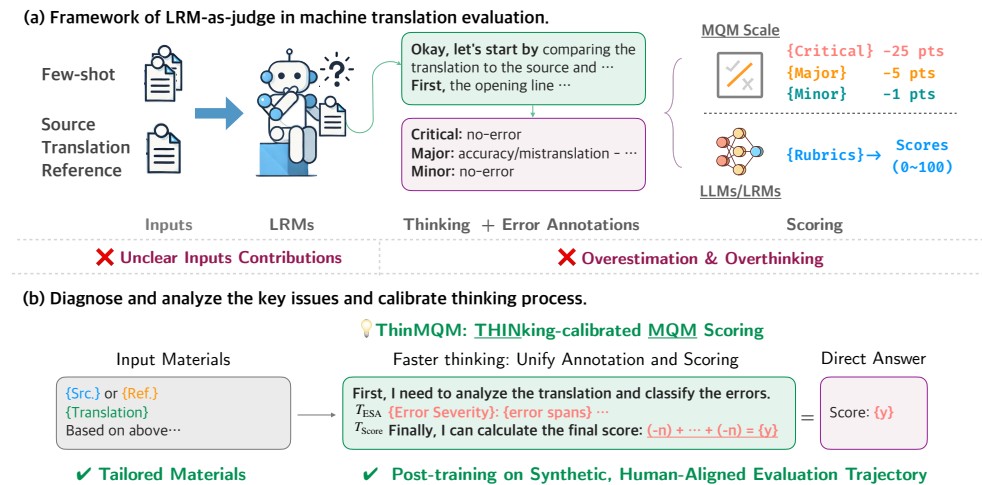

**Figure 2:** Overview of our research framework. (a) We decompose the general LRMs-as-judge pipeline for MT evaluation and identify key issues. (b) Guided by analytical findings, we propose ThinMQM, which establishes a more effective reasoning process.

the reference-free setting, evaluation is performed by comparing the hypothesis with the source text (*Src.*), i.e., $X_{\text{Src.}} = (h, s)$. Additionally, it is also possible to use all three components jointly (*Joint.*), i.e., $X_{\text{Joint.}} = (h, s, r)$, for scoring. In contrast, the emerging LLM-as-a-judge paradigm does not strictly conform to an end-to-end $X \rightarrow y$ paradigm, as it is not a parameterized regression model but rather a generative language model. Although such models can provide more detailed output information, such as explanations for the given scores, in most cases the desired score $y$ must be extracted post hoc from the model outputs using rule-based extractor or auxiliary scoring models.

### 2.2 MQM Framework and Related Work

Even we can list all the possible combinations of evaluation materials, modeling detailed scoring process remains a challenge. Earlier automated metrics focused on providing a single score value [6, 30], but they lacked consideration or transparency in aligning with human evaluation process, especially for neural metrics. Previous practices in collecting human annotations centered around direct assessment scores [31, 32], neglecting to establish a fine-grained, unified framework for score annotation. MQM is a professional scoring framework for translation quality assessment, designed to guide multi-dimensional evaluation of translations, including aspects such as fluency, terminology, and style [17, 22]. Typically, human annotators perform error span annotations, assign severity levels (e.g., major or minor), and finally, the score is aggregated by certain severity weights. In most cases, a major error incurs a penalty of -5 points, while a minor error results in a deduction of -1 point (or -0.1 for fluency errors). Critical errors, such as non-translation, are generally penalized by -25 points. The total score for an instance is computed by aggregating the penalties of all identified errors.

To develop more fine-grained machine translation evaluation metrics, recent research has focused on constructing automated methods aligned with MQM scoring. This has emerged as the primary evaluation task in the recent WMT leaderboard [33, 25]. The end-to-end approach [34, 9, 35, 29] essentially constructed upon pre-trained language models by finetuning on MQM scoring data. The LLM-as-judge approach to perform MQM scoring largely mirrors human annotation procedures: first, the model is instructed with MQM guidelines, and then it either extracts error span annotations to compute a score, or directly outputs a quality score. Currently, one of the most widely adopted and effective methods is the GEMBA series [13, 14] based on prompting GPT models. Its GEMBA-MQM variant further leverages in-context learning (ICL) [36, 37] by using three-shot demonstrations to assist in the evaluation process. We adapt it to LRM scope and conduct analysis in this paper.

## 3 Understanding LRM Behaviors in MT Evaluation

Figure 2 illustrates our research framework. In this section, we aim to address the first two research questions: how well LRM performs in MT evaluation and what failures occur in the practice.

## 3.1 Experimental Setup

**Methodology** As introduced earlier, following the successful SOTA practice in LLMs, we replicate GEMBA-MQM methodology on LRM as the basis for our analysis. We first instruct the LRM to annotate error spans in the translation according to the MQM framework, categorizing them into critical, major, and minor errors. Based on these error spans, we then apply a rule-based scoring mechanism to compute the final score for each translation. The penalty scheme for each type of error follows the approach described in Section 2.2.

Building on this, we conduct experiments with all possible input material combinations $X_{\text{Src.}\vee\text{Ref.}\vee\text{Joint.}}$ to investigate the influence of evaluation materials in Section 3.2. Meanwhile, the demonstrations used in ICL follow the same format as those in the GEMBA-MQM. It is worth noting that, since the MQM variant in the GEMBA series is reference-free, for reference-based setups (i.e., Ref. and Joint.), we supplement the demonstrations with reference information and adjust the prompt templates accordingly. Detailed prompts are provided in the Appendix C.2. We report the main results based on rule-based scoring mechanism and will discuss alternative model-based scoring methods (the logic is the same as the ESA prompting variant of GEMBA) in Section 3.3.

**Models Setups** We employ several mainstream LRM models across different sizes and architectures, including Deepseek-R1 671B, QwQ 32B, as well as distilled variants of R1 trained via knowledge distillation: R1-Distill-LLaMA 8B and R1-Distill-Qwen 7B. These models have demonstrated strong performance on complex reasoning tasks. Due to computational constraints, we are unable to deploy the open-source version of the R1 model locally and instead access it via API for experiments. All other models are deployed using the vLLM framework [2]. The decoding parameters are set as follows: temperature is set to 0.6, $\text{top}_p$ and $\text{top}_k$ is set to 0.95, 20. Our selection of DeepSeek-R1 is driven by its transparency of reasoning trajectories. In contrast, other frontier models, such as o3 and Gemini-2.5 Pro, do not expose their internal reasoning processes. This limitation makes them unsuitable for fair and fine-grained analysis. Nevertheless, we report the evaluation performance of Gemini-2.5 Pro in Section 4.2, but exclude it from the analytical sections.

**Data** We chose the WMT24 Metrics Shared Task [24] for our evaluation data in order to prevent potential issues with data contamination [38, 39, 40]. We confirmed that the release date for the WMT24 MQM data is after the knowledge cutoff for the models aforementioned. This task involves assessing the correlation between the evaluation models' scores and the human expert MQM scores, both at the system and segment levels. The WMT24 Metrics task includes three language pairs: English-German (En-De), English-Spanish (En-Es), and Japanese-Chinese (Ja-Zh), with around 20 machine translation systems for each pair.

**Meta-Evaluation Metrics** We used the same meta-evaluation settings as WMT24 official, focusing on key metrics such as system-level soft pairwise accuracy [41] (SPA) and tie-calibrated segment-level pairwise accuracy [42] ($Acc_{eq}^*$). Specfically, for SPA, it can be formally expressed as:

$$SPA = \binom{N}{2}^{-1} \sum_{i=0}^{N-1} \sum_{j=i+1}^{N-1} \left(1 - \left|p_{ij}^h - p_{ij}^m\right|\right) \tag{1}$$

where $N$ is the number of systems. $p_{ij}^h$ is the $p$-value that system $i$ is better than system $j$ based on human judgments, and $p_{ij}^m$ is the same based on metric scores. $\binom{N}{2}^{-1}$ normalizes over all pairs. Higher values of these metrics indicates stronger agreement between human and metric rankings. We used the MTME[3] to maintain consistency with the official calculations. We also report permutation-based significance testing with 1,000 resampling times [42] in comparison experiments.

## 3.2 Impact of Evaluation Materials

**Contribution Quantification** In MT evaluation, since the translation hypothesis $h$ is present in all evaluation scenarios, it is necessary to assess the impact of source and reference information on evaluation performance. While it is feasible to enumerate and experiment with all combinations of

---

[2]https://github.com/vllm-project/vllm
[3]https://github.com/google-research/mt-metrics-eval

evaluation materials, denoted as $X_{\text{Src.}\vee\text{Ref.}\vee\text{Joint.}}$, quantitatively isolating the contribution of each component remains challenging due to the overlapping presence of source and reference across multiple evaluation settings. To address this, we adopt the Shapley Value [43] as a principled measure to quantify the individual contributions of source and reference information to evaluation outcomes across different models and evaluation settings.

Formally, the Shapley Value of the source information $\phi_s$ is defined as:

$$\phi_s = \sum_{s' \subseteq N \setminus \{s\}} \frac{|s'|!(|N| - |s'| - 1)! \cdot (v(s' \cup \{s\}) - v(s'))}{|N|!} \tag{2}$$

where $N = \{s, r\}$ denotes the set of all materials that may affect evaluation (source $s$ and reference $r$). The translation hypothesis $h$ is always present and thus not part of the combinatorial set. The function $v(\cdot)$ represents the evaluation performance under a specific evaluation setting, which we quantify using system-level and segment-level metrics. The set $s'$ refers to all subsets of $N$ excluding $s$, i.e., $\{\emptyset, r\}$. In particular, the case $\emptyset$ corresponds to an evaluation setting with neither source nor reference (i.e., translation-only). $v(h)$ is a invalid value as translation-only is not a valid input, thus we only approximate it using an available configuration, namely $v(\{h, r\})$, to estimate $v(h)$. Therefore, taking into account the practical constraints of machine translation evaluation, we refer the approximated Shapley Value here as $\phi_s^{\text{MT}}$ in order to distinguish from the strict definition in Eq.2. The calculation of $\phi_r^{\text{MT}}$ follows analogously.

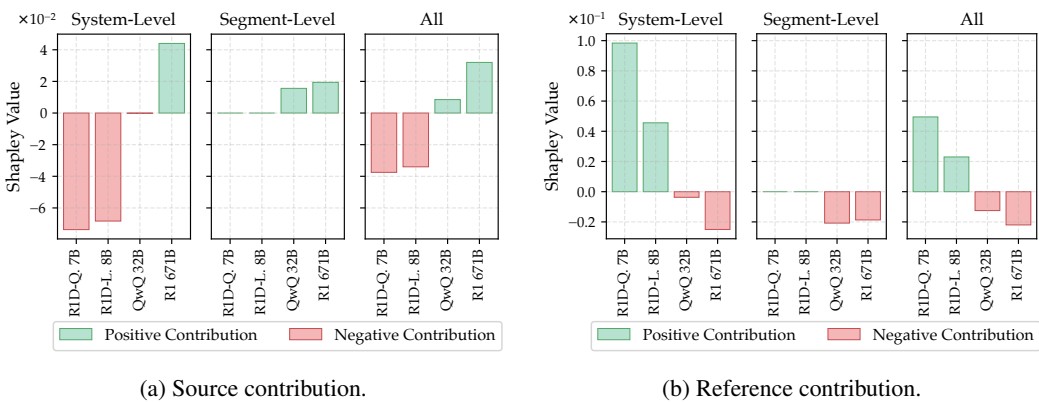

(a) Source contribution.     (b) Reference contribution.

Figure 3: Shapley value analysis of input contributions of evaluation materials at different evaluation granularities. "R1D-L./Q." refers to the R1-Distill-Llama/-Qwen models.

**Results and Discussion**  The results presented in Table 1 along with the significance tests, intuitively suggest that using either source or reference information as evaluation materials represents the most effective choice for LRM-based evaluation. However, this observation is contingent on model scale. The illustrative results shown in Figure 3, derived from Shapley Value $\phi^{\text{MT}}$ analysis, reveal a pattern: for smaller-scale LRMs (7/8B), source information is detrimental to evaluation quality, whereas reference information contributes positively. This trend is reversed in larger LRMs, such as QwQ 32B and R1 671B, where source information becomes beneficial and reference information less so.

Previous work [44] on the LLM-as-a-judge observed that LLMs tend to become "lost in source" during MT evaluation, regardless of model size. However, our findings suggest that this phenomenon may not generalize to the LRM-as-a-judge setting. This distinction is plausible, given that LRMs typically have stronger reasoning capabilities on complex tasks [45, 19, 20] compared to general-purpose LLMs. Earlier LLMs may have lacked the capacity to effectively model the cross-lingual relationships between source and translation.

As for the observed adverse impact of reference information on LRM-based evaluation, we hypothesize two possible factors. First, MQM human annotations are inherently reference-free, focusing solely on the source and translation without relying on references. Second, the quality of the reference itself significantly affects the correlation of automatic metrics with human judgment, as prior work [46, 33] has pointed out, "*BLEU (or Metrics) might be guilty, but reference not innocent*". These findings highlight the need for scale-aware design evaluation setups: the choice of evaluation materials should be informed by the capabilities and limitations of the model size in question.

Table 1: Comparison of different evaluation material setups. The highest value in each model is **bolded**, and † indicates results that are significantly better ($p < 0.05$) based on permutation tests.

| Materials | En-De SPA (%) | En-De $Acc^*_{eq}$ | En-Es SPA (%) | En-Es $Acc^*_{eq}$ | Ja-Zh SPA (%) | Ja-Zh $Acc^*_{eq}$ | Avg. SPA (%) | Avg. $Acc^*_{eq}$ | All |
|---|---|---|---|---|---|---|---|---|---|
| | | | | Deepseek-R1 671B | | | | | |
| *Src.* | 82.1 | **47.4**† | **77.8** | 68.0 | 90.4 | **46.8**† | **83.4** | **54.1** | **68.8**† |
| *Ref.* | 80.4 | 43.0 | 67.3 | 68.0 | 88.6 | 43.6 | 78.8 | 51.5 | 65.2 |
| *Joint.* | **82.4** | 46.6 | 75.6 | 68.0 | **91.1** | 44.0 | 83.0 | 52.9 | 68.0 |
| | | | | QwQ 32B | | | | | |
| *Src.* | 79.8 | **46.8**† | **76.1**† | 68.0 | 91.9 | **46.9**† | **82.6** | **53.9** | **68.3**† |
| *Ref.* | **84.2** | 42.9 | 68.7 | 68.0 | **94.3** | 43.5 | 82.4 | 51.5 | 66.9 |
| *Joint.* | 81.7 | 44.2 | 72.2 | 68.0 | 92.5 | 44.3 | 82.1 | 52.2 | 67.2 |
| | | | | R1-Distill-Llama 8B | | | | | |
| *Src.* | 72.3 | 42.9 | 65.9 | 68.0 | 74.2 | 43.5 | 70.8 | 51.5 | 61.1 |
| *Ref.* | 71.8 | 42.9 | **78.5**† | 68.0 | **84.7** | 43.5 | **78.3** | 51.5 | **64.9**† |
| *Joint.* | **72.9** | 42.9 | 65.1 | 68.0 | 78.7 | 43.5 | 72.2 | 51.5 | 61.9 |
| | | | | R1-Distill-Qwen 7B | | | | | |
| *Src.* | 52.6 | 42.9 | 53.9 | 68.0 | 68.3 | 43.5 | 58.3 | 51.5 | 54.9 |
| *Ref.* | **67.3**† | 42.9 | **61.0** | 68.0 | 83.8 | 43.5 | **70.7** | 51.5 | **61.1** |
| *Joint.* | 58.4 | 42.9 | 57.0 | 68.0 | **86.1** | 43.5 | 67.2 | 51.5 | 59.3 |

## 3.3 Pitfalls of Scoring Mechanisms

Table 2: Effect of changing the rule-based scoring weights on average correlation Avg. Δ ($Acc^*_{eq}$., SPA) metrics.

| Model | Src. | Joint. | Ref. | Avg. Δ |
|---|---|---|---|---|
| R1 671B | +0.60 | +0.60 | -0.50 | +0.23 |
| QwQ 32B | +0.20 | +0.50 | +0.30 | +0.33 |
| R1D-L.8B | -1.20 | -0.20 | -0.80 | -0.73 |
| R1D-Q.7B | -0.10 | -0.30 | -1.10 | -0.50 |
| **Avg. Δ** | -0.13 | +0.15 | -0.53 | – |

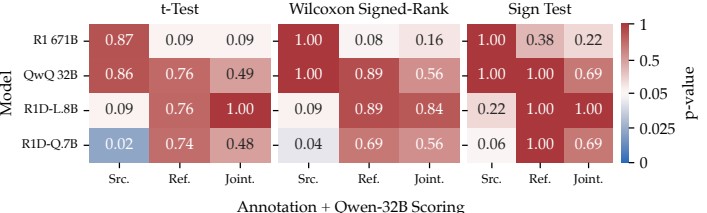

Figure 4: Significance testing of the contribution of the auxiliary scoring model (Qwen-2.5 32B). Test results ($p < 0.05$) are highlighted in distinct colors and scales.

**Dilemma in Post-Scoring** Since LRM-based evaluators generate descriptive outputs in an autoregressive manner, an additional critical factor influencing scoring is how error spans are extracted and scored. In this work, we consider two commonly adopted paradigms: rule-based and model-based re-scoring. The rule-based scoring introduced before aggregate scores under MQM standard, offering transparency. In contrast, the model-based scoring paradigm allows for an additional stage of reflection but lacks transparency. Naturally, for a rule-based scorer, the robustness to rule changes is worth noticing. Furthermore, when auxiliary models are used for scoring, it becomes difficult to disentangle whether observed improvements stem from the LRM or auxiliary scorer.

We first investigate using the LRM itself to score error spans annotated via the GEMBA-ESA protocol. The results show no improvement over the rule-based scorer; in fact, we observe a slight performance drop (e.g., QwQ 32B yields a mean performance 68.3 → 68.1), along with significantly higher inference costs. Next, we employ an auxiliary model (Qwen-2.5 32B) to perform the same scoring procedure. In this setting, we find that the meta-evaluation results of LRM closely align with the that of the auxiliary model itself. This raises a critical question: are the observed gains attributable to the LRM outputs, or simply to the auxiliary model? To address this, we conduct statistical significance testing and in-depth comparison of score distributions.

**Results and Discussions** To answer above question, we perform significance testing between the scores produced by the "LRM + auxiliary" setup and those from the auxiliary model alone. The results in Figure 4 demonstrate that the re-scoring process using an external model fails to provide clear attribution regarding the source of evaluation performance. This finding highlights the need to either

enhance the scoring capabilities of the original LRM or adopt transparent, rule-based extraction and scoring mechanisms. Figure 5 further compares the distributions of two scoring paradigms.

The results reveal a persistent overestimation issue in model-based evaluation. The instances that human annotators consider error-free are still judged as erroneous by the LRM. This indirectly confirms that the improvements brought by auxiliary models do not fundamentally resolve the shortcomings of LRM-based scoring and are insufficient to mitigate the overestimation problem, requiring for a more human-centric evaluation process that mirrors human judgment rules for better correlations.

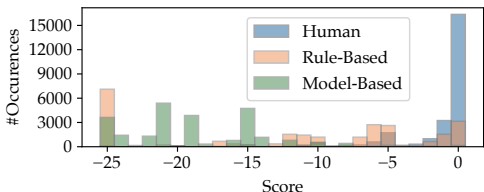

Figure 5: Distribution of evaluation scores across different scoring paradigms.

Another remaining concern with a rule-based scorer is how sensitive the results are to the choice of scoring scheme. To study this, we conducted an experiment using an alternative severity weighting scheme (i.e., -3/-2/-1). Table 2 reports the average change $\Delta$ across all correlation metrics. We observe that although adjusting the weights does slightly shift the absolute correlation values, the differences are modest. A likely explanation is that meta-evaluation metrics are primarily sensitive to the rank order of segments. As long as the ordinal structure of the penalties is preserved, the rankings remain relatively stable, supporting the robustness of the rule-based scoring approach.

### 3.4 "Overthinking" Process: When More is Not Better

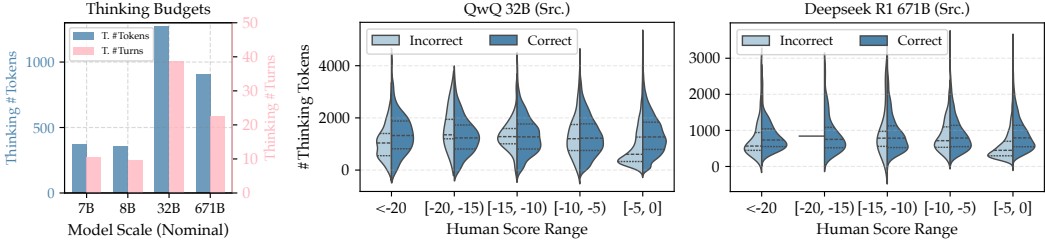

(a) Budget statics across all languages.          (b) Budget allocations across varying evaluation difficulties.

Figure 6: Analysis of thinking budget attribution across model scale and evaluation difficulty. Appendix B.4 includes all the results under different settings.

**Thinking Budgets**  LRMs typically benefit from scaling test-time thinking budget. Intuitively, we investigate whether such scaling is both effective and efficient in MT evaluation. We quantify the thinking budget along two dimensions: 1) the number of tokens generated during the reasoning process, and 2) the number of reasoning turns[4]. Additionally, since two of the used LRMs also have corresponding general-purpose LLMs, we also examine their performance to assess whether LRM post-training contributes to improved performance[5].

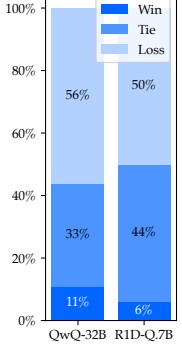

**Results and Discussions**  The results in Figure 6(a) show that reasoning cost is not necessarily tied to model size. For example, QwQ, despite not being the most powerful LRM-as-a-judge model, incurs the highest reasoning cost.

Moreover, the thinking budget is also unrelated to instance difficulty. As shown in Figure 6(b), median thinking tokens remain stable across difficulty levels. An exception appears at the extremes of evaluation difficulty, where human scores are either very high or very low. The correctly aligned (i.e., model scores consistent with human judgment) predictions require more effort, while misaligned ones are cheaper. This is the only case where the thinking budget shows a hint of rational allocation.

Figure 7: Comparison between LRM and its corresponding LLM.

---

[4]Empirically, we observe that these LRMs use output delimiter across reasoning turns.

[5]We exclude R1-Distill-Qwen 7B from this comparison, as its original model is not general-purpose LLM.

On the other hand, we compute meta-evaluation metrics across different languages and evaluation levels, comparing LRMs with their corresponding general-purpose LLMs. Statistical significance testing is used to categorize outcomes as wins, ties, or losses, shown in Figure 7. "Tie" is considered as failed significance test. The results show that LRMs underperform in nearly half of the evaluation settings, indicating that current general-purpose LRMs, despite their "slow-thinking" process, still struggle to consistently enhance evaluation performance.

## 4 Improving LRM via Human-Aligned Thinking Trajectory

### 4.1 Methodology

**Motivations** Our analysis reveals that standard practices of LLMs in MT evaluation are not universally optimal for LRMs. The unconstrained "thinking" processes within LRMs can be inefficient and may lead to overestimated score. Key insights derived from this analysis include: 1) Reference-free evaluation is preferable for strong LRMs, while reference-based evaluation remains suitable for weaker LRMs. 2) Aligning the LRM's reasoning process with specific scoring rubrics is crucial. 3) Extensive budget allocated to LRM "thinking" do not consistently improve performance.

**ThinMQM** Drawing from previous observations, we introduce **Thin**king-calibrated **MQM** (Thin-MQM) scoring method, a methodology designed to adapt LRMs to emulate human evaluation process. The core idea is intuitive: generating synthetic data that mirrors the human MQM workflow, thereby calibrating and aligning the LRM's internal thinking process with the pipeline of human evaluation.

Given human MQM annotations which consist of error spans $E = \{e_1, e_2, \ldots, e_n\}$ and their associated severity levels $L = \{l_1, l_2, \ldots, l_n\}$, we aim to model the human two-phase evaluation process. This process involves an initial error span annotation (ESA) stage, $T_{\text{ESA}} : X \rightarrow (E, S)$, followed by a scoring stage based on a rubric, $T_{\text{score}} : (E, S) \rightarrow \text{Score}_{MQM}$. We transform this sequence into a concise yet effective structured thinking chain, intended to serve as a proxy for human annotation steps. The resulting synthesized data, $\mathcal{D}_{\text{synth}} = \{(X_{\text{Src.}\vee\text{Ref.}}, [T_{\text{ESA}}(X), T_{\text{score}}(T_{\text{ESA}}(X))])\}$, adheres to the structure shown in Figure 2 (b). The LRM, denoted by $M_\theta$, would be post-trained on the dataset $\mathcal{D}_{\text{synth}}$, with parameters $\theta$ producing the output sequences. The fine-tuning process seeks to update $\theta$ to $\theta'$ by minimizing the cross-entropy loss function $\mathcal{L}_{\text{CE}}$ over all instances in $\mathcal{D}_{\text{synth}}$:

$$\theta' \leftarrow \arg\min_\theta \sum_{\mathcal{D}_{\text{synth}}} \mathcal{L}_{\text{CE}}(M(X_{\text{Src.}\vee\text{Ref.}}; \theta), [T_{\text{ESA}}(X), T_{\text{score}}(T_{\text{ESA}}(X))]) \tag{3}$$

### 4.2 Experiments

**Data** We synthesized ThinMQM training data based on the human-annotated MQM dataset from WMT23, which includes two evaluation tasks: English–German and Chinese-English. Synthetic data instances were constructed based on the methodology described above and the prompt templates detailed in Appendix C.4. Due to an imbalance distribution across the two language pairs, we down-sampled the larger set to ensure balanced training data. The final dataset consists of approximately 5,980 instances per language pair, yielding a total of 11,960 training instances.

**Model and Setups** To verify the effectiveness of ThinMQM on various model sizes, we fine-tune 7B, 8B, and 32B models. Based on earlier analysis, we adopt a reference-based evaluation setup (Ref.) for the 7B and 8B models in both training and inference, while the 32B model employed a reference-free setup (Src.). All models are fine-tuned for 4 epochs with a learning rate of $1e - 5$, and the total batch size is 32. Other training hyper-parameters are detailed in Appendix C.1.

**Main Results** The results in Table 3 clearly demonstrate that post-training calibration with Thin-MQM significantly improves LRM performance under the same evaluation setups. Specifically, the 7B model shows gains of up to +8.7 points in meta-evaluation metrics, while the 32B model achieves a +3.9 points improvement, reaching performance comparable to state-of-the-art metrics such as xCOMET, despite those relying on training on large-scale MQM data and have different model architectures. Notably, within the LLM/LRM evaluation paradigm, our ThinMQM-32B model achieves superior average performance compared to the baselines, though not necessarily on every individual language pair.

Table 3: Performance comparison of different models. The highest value is **bolded**, and the second-best is underlined. † denotes significantly better ($p < 0.05$) results based on permutation tests.

| Metric/Model | Avg. All | En-De SPA (%) | En-De $Acc^*_{eq}$ | En-Es SPA (%) | En-Es $Acc^*_{eq}$ | Ja-Zh SPA (%) | Ja-Zh $Acc^*_{eq}$ |
|---|---|---|---|---|---|---|---|
| BLEU [6] | 58.9 | 73.7 | 43.1 | 51.4 | 68.0 | 73.6 | 43.5 |
| COMET-22 [9] | 68.9 | 87.9 | 48.2 | 77.9 | 68.3 | 81.4 | 49.6 |
| xCOMET [10] | 71.9 | **90.6** | **53.0**$^†$ | 78.9 | 68.8 | 88.9 | 51.0 |
| GEMBA-ESA [13] | 71.1 | 79.1 | 50.7 | **84.0** | 68.3 | 90.8 | 53.9 |
| Gemini-2.5-Pro [47] | 71.0 | 82.3 | 51.2 | 76.9 | 68.0 | **94.8** | 53.1 |
| Deepseek-R1 [19] | 68.8 | 82.1 | 47.4 | 77.8 | 68.0 | 90.4 | 46.8 |
| QwQ 32B | 68.3 | 79.8 | 46.8 | 76.1 | 68.0 | 91.9 | 46.9 |
| + *ThinMQM* | **72.2**$_{+3.9}$ | 83.2$_{+3.4}$ | 52.5$_{+5.7}$ | 80.7$_{+4.6}$ | **69.2**$^†_{+1.2}$ | 91.3$_{-0.6}$ | **56.1**$^†_{+9.2}$ |
| R1-Distill-Llama-8B | 64.9 | 71.8 | 42.9 | 78.5 | 68.0 | 84.7 | 43.5 |
| + *ThinMQM* | 70.8$_{+5.9}$ | 85.5$_{+13.7}$ | 48.6$_{+5.7}$ | 81.3$_{+2.8}$ | 68.2$_{+0.2}$ | 90.5$_{+5.8}$ | 51.0$_{+7.5}$ |
| R1-Distill-Qwen-7B | 61.1 | 67.3 | 42.9 | 61.0 | 68.0 | 83.8 | 43.5 |
| + *ThinMQM* | 69.8$_{+8.7}$ | 84.5$_{+17.2}$ | 48.5$_{+5.6}$ | 77.8$_{+16.8}$ | 68.0$_{+0.0}$ | 89.0$_{+5.2}$ | 51.3$_{+7.8}$ |

## 4.3 Analysis

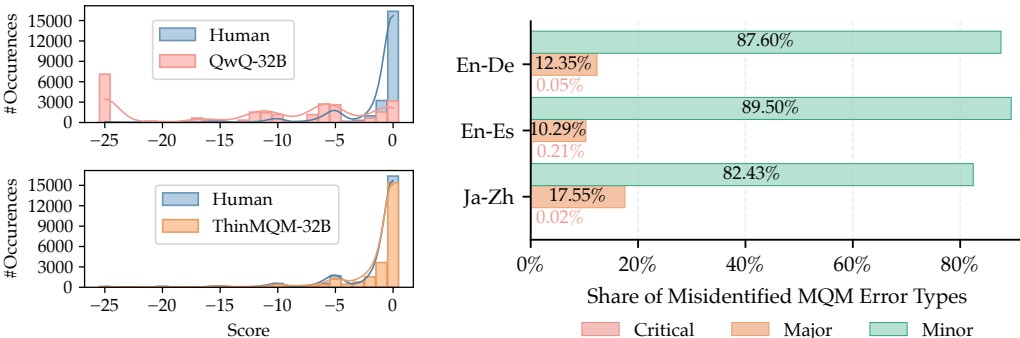

Figure 8: Comparison of scoring distributions between ThinMQM and QwQ-32B.

Figure 9: Distribution of ThinMQM-32B–human judgment discrepancies across MQM error types.

**Scoring Distribution**  As shown in Figure 8, the primary reason behind the improvement brought by ThinMQM lies in the calibrated scoring distribution[6]. Specifically, ThinMQM effectively mitigates the overestimation problem, aligning the model predicted scores more closely with the human MQM distribution, particularly in cases with non-error cases. This finding echoes the earlier observations in Figure 5, indicating that the performance gains from ThinMQM are both meaningful and justified.

**Error Typology**  To analyze cases where ThinMQM-32B diverges from human judgments, we categorize these discrepancies according to the MQM translation error taxonomy (Critical, Major, Minor), as shown in Figure 9. The analysis shows that the largest misalignment arises from Minor-level errors. Moreover, within the Minor category, *accuracy/mistranslation* accounts for the highest proportion of discrepancies, highlighting areas where future improvements should be targeted.

**Efficiency**  As illustrated in Figure 1 (b), ThinMQM reduced the unnecessary thinking budget while maintaining high evaluation performance. This indicates that post-training alignment not only improves effectiveness but also enhances efficiency. In practice, This represents a substantial decrease in the computational cost of LRM-based translation evaluation. For example, when evaluating English–German translations with QwQ 32B under the vLLM framework using four A100 GPUs, inference time is reduced from 12 minutes per 1,000 examples to 40 seconds on average.

---

[6]Please refer to Appendix B.1 for detailed language-specific distributions due to space limitation.

Table 4: Multi-run evaluation of ThinMQM at temperature 0.6. Each score is presented as $\text{Mean}_{\text{Std.}}$ over 3 independent runs.

| | Avg. | En-De | | En-Es | | Ja-Zh | |
|---|---|---|---|---|---|---|---|
| Model | All | SPA (%) | $Acc^*_{eq}$ | SPA (%) | $Acc^*_{eq}$ | SPA (%) | $Acc^*_{eq}$ |
| ThinMQM 32B | $72.0_{.003}$ | $80.7_{.026}$ | $53.1_{.006}$ | $80.8_{.002}$ | $68.9_{.002}$ | $92.5_{.011}$ | $55.9_{.002}$ |
| ThinMQM 8B | $70.4_{.004}$ | $83.1_{.021}$ | $48.6_{.003}$ | $81.3_{.020}$ | $68.2_{.001}$ | $90.3_{.013}$ | $51.0_{.001}$ |
| ThinMQM 7B | $70.0_{.002}$ | $85.4_{.011}$ | $48.2_{.004}$ | $77.6_{.002}$ | $68.1_{.001}$ | $89.3_{.004}$ | $51.4_{.003}$ |

Table 5: Performance comparison in Hindi-Chinese MQM.

| Model | Sys. $\rho$ | Seg. $\tau$ |
|---|---|---|
| XCOMET-XXL | 62.5 | 47.8 |
| ThinMQM 32B | **63.4** | **57.4** |
| ThinMQM 7B | 51.3 | 49.1 |
| ThinMQM 8B | 50.3 | 47.5 |

## 4.4 Ablation Study

**Stability**  Figure 10 demonstrates that ThinMQM is robust to test-time hyperparameter choices. We evaluate performance under various decoding settings and compare meta-metrics' scores. For example, QwQ-32B's system-level evaluation is sensitive under greedy decoding, whereas ThinMQM remains stable, nit with only a drop at the segment level. To further verify stability, we conduct three runs at a fixed temperature of 0.6. As shown in Table 4, the low standard deviation confirms that ThinMQM's performance is consistent and not subject to significant random fluctuations. Overall, these results validate our chosen decoding configuration as a broadly robust setup for LRM-based evaluation.

**Generalization**  To perform a more stringent out-of-distribution test on a low-resource language pair, we sourced a recently released Hindi-Chinese dataset with MQM annotations [48], which was published after our LRMs' knowledge cutoff date. Since this dataset contains translations from fewer than four systems, we use system-level Pearson $\rho$ and Kendall correlation $\tau$ as meta-evaluation metrics. As shown in Table 5, ThinMQM demonstrates generalization capabilities under low-resource scenarios, outperforming the xCOMET-XXL baseline.

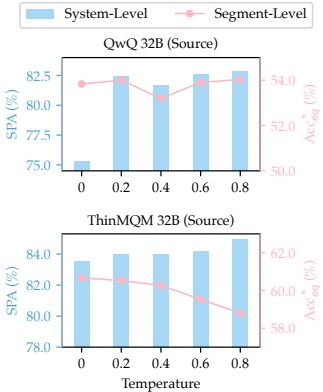

Figure 10: Performance of different models under varying temperature setups.

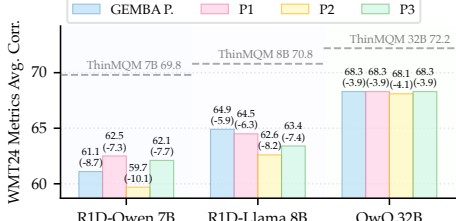

Figure 11: Comparison of ThinMQM with baselines using paraphrased prompts.

**Prompting Templates**  To further strengthen the comparison between our proposed ThinMQM and alternative baseline prompting templates, we used GPT-4o to paraphrase our GEMBA-MQM prompt and generated three additional variants, which we denote as P1-P3. Figure 11 (details in Section B.3) shows that ThinMQM consistently maintains a performance advantage across model sizes. Besides, additional interesting observations emerge from these results. For LRM baselines, large models (32B) are relatively insensitive to prompt variation, exhibiting only minimal differences in performance. In contrast, smaller models (7-8B) are more sensitive to prompts. However, the resulting fluctuations are limited and still do not surpass the ThinMQM.

## 5 Conclusions

In this paper, we presented a systematic investigation into LRM-as-a-judge for machine translation evaluation, exploring their capacity to model the process of MQM assessment task. Our analysis across various LRMs revealed that there is a need to tailor evaluation materials for evaluation and they "overthink" simple instances, exhibiting overestimation biases. To address this, we introduced a simple yet effective method of calibrating LRM thinking by training them on synthetic, human-like MQM evaluation trajectories. This approach substantially reduced thinking budgets while improving evaluation performance on WMT24 Metrics benchmarks, primarily by calibrating scoring distributions and reducing overestimation. Our findings demonstrate the potential of LRMs for MT evaluation but highlight the critical need for controlled thinking and careful calibration to realize their full potential in translation evaluation, paving the way for future advancements in developing better LRM-as-a-judge in MT evaluation. Future work will extend evaluation to more diverse languages.

## Acknowledgments

This work was supported in part by the Science and Technology Development Fund of Macau SAR (Grant No. FDCT/0070/2022/AMJ, China Strategic Scientific and Technological Innovation Cooperation Project Grant No. 2022YFE0204900), the Science and Technology Development Fund of Macau SAR (Grant No. FDCT/0007/2024/AKP), the Science and Technology Development Fund of Macau SAR (Grant No. FDCT/060/2022/AFJ, National Natural Science Foundation of China Grant No. 62261160648), the UM and UMDF (Grant Nos. MYRG-GRG2023-00006-FST-UMDF, MYRG-GRG2024-00165-FST-UMDF, EF2023-00151-FST, EF2023-00090-FST, EF2024-00185-FST), and the National Natural Science Foundation of China (Grant No. 62266013). We would like to thank the anonymous reviewers for their insightful comments.

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

# Appendix

## A  Limitations

Our proposed calibration method relies on synthetic thinking trajectories designed to be "human-like" and "findings-driven". However, these synthetic datasets may not fully capture the diversity of human cognitive processes during MT evaluation. Additionally, since WMT24 includes MQM human ratings for only three language pairs, previous benchmarks have faced risks of data contamination. Our evaluations were primarily conducted using the WMT24 benchmarks, which may not represent all language pairs or domains equally. Lastly, this work follows an "understanding and then improving" approach. While we focus on analyzing the behavior of the LRM, the current calibration method primarily targets the efficiency of the thinking process (reducing "overthinking") and calibrating scoring distributions. More nuanced aspects of the reasoning process, such as the LRM's ability to consistently identify specific error types with fine granularity, may require more targeted or advanced alignment techniques.

## B  Supplementary Details

### B.1  Language-specific Scoring Distributions

Figure 12 presents all the scoring distributions when evaluating instances of different languages.

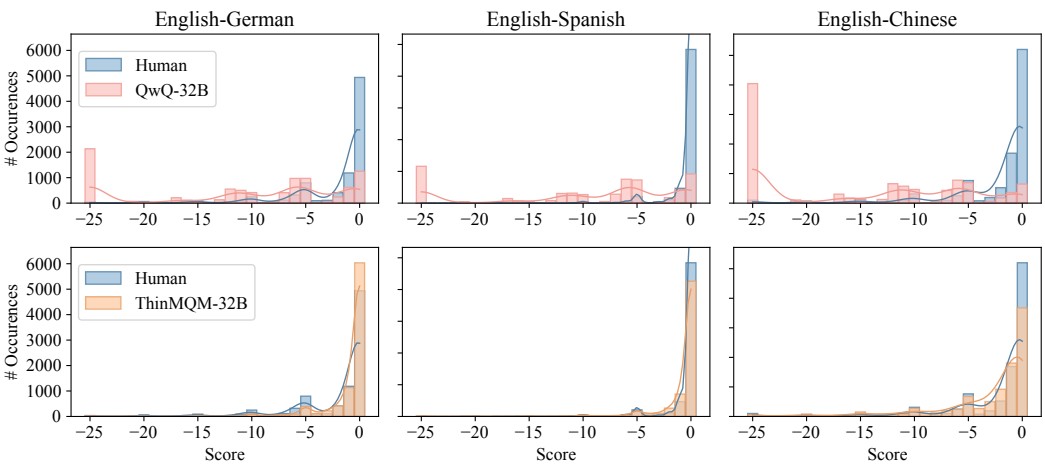

Figure 12: Distribution of MQM scores for QwQ-32B (top row) and ThinMQM-32B (bottom row) compared to human evaluations across different language pairs.

### B.2  Data Contamination Prevention

To ensure the integrity of our evaluation and prevent data contamination, we implemented the following rigorous measures. For all LRMs evaluated, we rigorously verified that their official knowledge cutoff date or public release date precedes the release date of our evaluation benchmarks. This chronological separation guarantees that the models were not exposed to the test data during their original training phase. Our synthetic training data, ThinMQM, is derived from the WMT23 dataset. The source data utilized for the synthesis of the synthetic set was finalized prior to the public release of the WMT24 MQM evaluation benchmark. This temporal order ensures no overlap between our training data and the final test set. To provide full transparency, we detail the relevant dates for all major components used in this study in Table 6.

### B.3  Supplementary Ablation Results

**Effect of ICL Demonstrations**  We further analyze the effects of ICL on the baseline model in Table 7 using QwQ-32B model. The results indicate that ICL is generally beneficial, improving the

Table 6: Knowledge cutoff and release dates for all models, data sources, and benchmarks used in this work. This chronology confirms the prevention of data contamination.

| Component | Type | Release Date / Knowledge Cutoff |
|---|---|---|
| WMT24 MQM | Evaluation Benchmark | *Oct 4, 2024* |
| Hindi-Chinese Expert MQM | Evaluation Benchmark | *Nov 26, 2024* |
| QwQ 32B | LRM | *Sep 19, 2024* (Qwen 2.5 Base) |
| R1-Distill-Llama 8B | LRM | *Dec, 2023* (Llama 3.1 Base) |
| R1-Distill-Qwen 7B | LRM | *Sep 19, 2024* (Qwen 2.5 Base) |
| WMT23 | Training Data Source | *Aug 10, 2023* |

Table 7: Effects of ICL. The highest value in each block is **bolded**.

| | En-De | | En-Es | | Ja-Zh | | Avg. | | |
|---|---|---|---|---|---|---|---|---|---|
| Model | SPA (%) | $Acc_{eq}^*$ | SPA (%) | $Acc_{eq}^*$ | SPA (%) | $Acc_{eq}^*$ | SPA (%) | $Acc_{eq}^*$ | All |
| | | | | | QwQ 32B | | | | |
| *Joint* | 81.7 | 44.2 | **72.2** | **68.0** | **92.5** | **44.3** | 67.2 | **82.1** | **52.2** |
| ↪ *w/o* ICL | **80.6** | **42.9** | 68.4 | 68.0 | 90.2 | 43.5 | 65.6 | 79.7 | 51.5 |
| *Src.* | 79.8 | 46.8 | 76.2 | 68.0 | 91.9 | **46.9** | 68.3 | **82.6** | **53.9** |
| ↪ *w/o* ICL | **78.0** | **44.8** | **76.5** | 68.0 | **92.6** | 45.7 | 67.6 | 82.4 | 52.8 |
| *Ref.* | **84.2** | 42.9 | **68.8** | 68.0 | **94.2** | 43.5 | 66.9 | **82.4** | 51.5 |
| ↪ *w/o* ICL | 74.8 | **42.9** | 64.0 | 68.0 | 90.0 | 43.5 | 63.9 | 76.3 | 51.5 |

average score for both the *Joint* and *Src.* settings, with the *Src.* variant achieving the best overall performance. The effect on the *Ref.* setting is more nuanced; while ICL significantly boosts the average SPA (%) (82.4 vs. 76.3), it has no discernible impact on the $Acc_{eq}^*$ score, resulting in an identical average score of 51.5 for both configurations. These results confirm that ICL is generally an effective strategy for improving overall model performance.

Table 8: Details of Training configuration.

| Hyperparameter | Value |
|---|---|
| Batch size per device | 2 for 7B/8B, 1 for 32B |
| Gradient accumulation steps | 4 |
| Learning rate | $1.0 \times 10^{-5}$ |
| Training epochs | 4.0 |
| Learning rate scheduler | cosine |
| Warmup ratio | 0.1 |
| Mixed precision (bfloat16) | Enabled |

**Detailed Results of Prompt Variation** As shown in Table 9, the ThinMQM model family still establishes a strong performance baseline when changing the prompt templates of baselines, outperforming all other tested model variants across all the scales. Our investigation into prompt sensitivity for the QwQ and R1-Distill models reveals inconsistent effects. For the QwQ 32B model, performance is relatively stable, with prompts P1 and P3 matching the GEMBA prompt baseline. Conversely, for the R1-Distill-Qwen 7B model, prompt P1 provides a notable improvement, boosting the average score from 61.1 to 62.5. Prompt P2, however, consistently degrades performance across all models. Most strikingly, for both the 8B and 7B models, the $Acc_{eq}^*$ scores remain completely static regardless of the prompt, suggesting that while prompt engineering can influence SPA (%), it fails to improve $Acc_{eq}^*$ for these models, supporting the choice of GEMBA prompting template.

Table 9: Performance comparison of baseline using different prompts. The highest value in each model is **bolded** and the second-best is underlined.

| Model | En-De SPA (%) | En-De $Acc^*_{eq}$ | En-Es SPA (%) | En-Es $Acc^*_{eq}$ | Ja-Zh SPA (%) | Ja-Zh $Acc^*_{eq}$ | Avg. SPA (%) | Avg. $Acc^*_{eq}$ | Avg. All |
|---|---|---|---|---|---|---|---|---|---|
| ThinMQM 32B | **83.2** | **52.5** | **80.7** | **69.2** | 91.3 | **56.1** | **85.1** | **59.3** | **72.2** |
| QwQ 32B (Gemba P.) | 79.8 | 46.8 | 76.1 | 68.0 | 91.9 | 46.9 | 82.6 | 53.9 | 68.3 |
| ↪ w/ P1 | 77.8 | 47.4 | 79.3 | 68.0 | 89.8 | 47.4 | 82.3 | 54.3 | 68.3 |
| ↪ w/ P2 | 77.0 | 46.9 | 74.4 | 68.0 | **93.7** | 48.4 | 81.7 | 54.4 | 68.1 |
| ↪ w/ P3 | 79.3 | 46.9 | 78.2 | 68.0 | 90.9 | 46.4 | 82.8 | 53.8 | 68.3 |
| ThinMQM 8B | **85.5** | **48.6** | **81.3** | **68.2** | **90.5** | **51.0** | **85.8** | **55.9** | **70.8** |
| R1D-L. 8B (Gemba P.) | 71.8 | 42.9 | 78.5 | 68.0 | 84.7 | 43.5 | 78.3 | 51.5 | 64.9 |
| ↪ w/ P1 | 74.5 | 42.9 | 72.3 | 68.0 | 85.7 | 43.5 | 77.5 | 51.5 | 64.5 |
| ↪ w/ P2 | 71.1 | 42.9 | 65.5 | 68.0 | 84.3 | 43.5 | 73.6 | 51.5 | 62.6 |
| ↪ w/ P3 | 70.2 | 42.9 | 72.2 | 68.0 | 83.3 | 43.5 | 75.2 | 51.5 | 63.4 |
| ThinMQM 7B | **84.5** | **48.5** | **77.8** | 68.0 | **89.0** | **51.3** | **83.8** | **55.9** | **69.8** |
| R1D-Q. 7B (Gemba P.) | 67.3 | 42.9 | 61.0 | 68.0 | 83.8 | 43.5 | 70.7 | 51.5 | 61.1 |
| ↪ w/ P1 | 70.8 | 42.9 | 70.0 | 68.0 | 79.6 | 43.5 | 73.5 | 51.5 | 62.5 |
| ↪ w/ P2 | 63.1 | 42.9 | 58.5 | 68.0 | 82.4 | 43.5 | 68.0 | 51.5 | 59.7 |
| ↪ w/ P3 | 66.5 | 42.9 | 69.6 | 68.0 | 81.9 | 43.5 | 72.7 | 51.5 | 62.1 |

## B.4 Scoring Distribution of Different Evaluation Difficulty

Figure 13 presents all the scoring distributions when evaluating instances at varying difficulty level.

# C Supplementary Setups

## C.1 Training Setups

We train the 7B/8B models using 4 A100 GPUs and the 32B model using 8 A100 GPUs. To enhance training efficiency, we utilize the DeepSpeed (Zero3) framework[7] for offloading. The settings are detailed in Table 8.

---

[7] https://github.com/deepspeedai/DeepSpeed

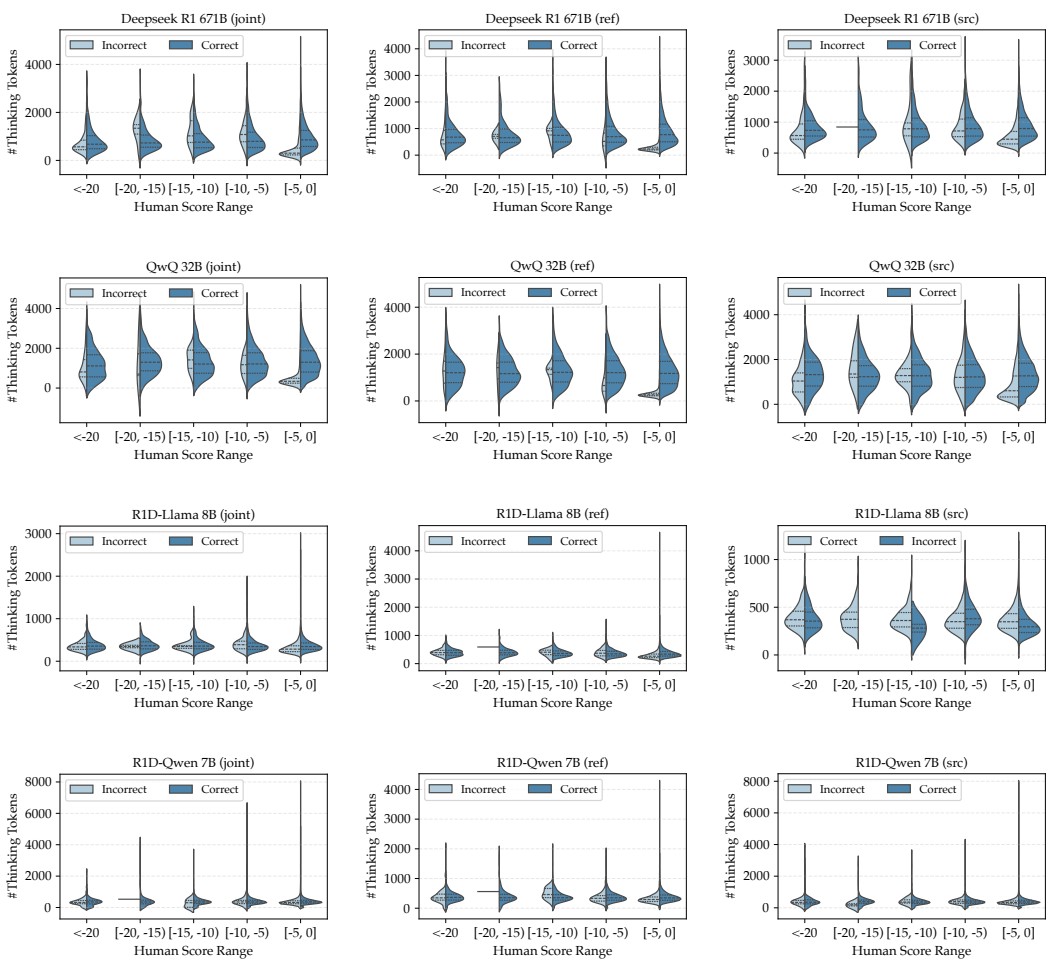

Figure 13: Distribution of thinking tokens under different evaluation difficulty.

## C.2 LRM-as-a-judge Prompt

For different prompting templates, we present only one specific evaluation setup as a demonstration. For the others, we simply remove or add some information.

---

**LRM-as-a-judge Prompt (Adapted from GEMBA-MQM, Joint. Setting)**

{Source Language} source:
{Source Text}
{Target Language} human reference:
{Reference Text}
{Source Language} translation:
{Translation Text}
Based on the source segment, human reference and machine translation surrounded with triple backticks, identify error types in the translation and classify them. The categories of errors are: accuracy (addition, mistranslation, omission, untranslated text), fluency (character encoding, grammar, inconsistency, punctuation, register, spelling), style (awkward), terminology (inappropriate for context, inconsistent use), non-translation, other, or no-error. Each error is classified as one of three categories: critical, major, and minor. Critical errors inhibit comprehension of the text. Major errors disrupt the flow, but what the text is trying to say is still understandable. Minor errors are technically errors, but do not disrupt the flow or hinder comprehension. Strictly output error classification results in this format:
Critical:
[error_type]-[error_spans] (one per line, use no-error if empty)
Major:
[error_type]-[error_spans] (one per line, use no-error if empty)
Minor:
[error_type]-[error_spans] (one per line, use no-error if empty).

---

## C.3 Auxiliary Scoring Model

---

**LLM Post-scoring Prompt (Adapted from GEMBA-ESA, Ref. Setting)**

Given the translation from {Source Language} to {Target Language} and the annotated error spans, assign a score on a continuous scale from 0 to 100. The scale has following reference points: 0="No meaning preserved", 33="Some meaning preserved", 66="Most meaning preserved and few grammar mistakes", up to 100="Perfect meaning and grammar".
Score the following translation from
{Source Language} source: "'{Source Text}'"
{Target Language} translation: "'{Translation Text}'"
Annotated error spans:
"'{ESA Details}'"
Score (0-100):

---

## C.4 ThinMQM Template

---

**ThinMQM Template (Src. Setting)**

**INPUT:**
{Source Language} source:
{Source Text}
{Target Language} translation:
{Translation Text}
Based on the source segment and machine translation surrounded with triple backticks, identify error types in the translation, classify them and give a score. The categories of errors are: accuracy (addition, mistranslation, omission, untranslated text), fluency (char-

---

acter encoding, grammar, inconsistency, punctuation, register, spelling), style (awkward), terminology (inappropriate for context, inconsistent use), non-translation, other, or no-error. Each error is classified as one of three categories: critical, major, and minor. Critical errors inhibit comprehension of the text. Major errors disrupt the flow, but what the text is trying to say is still understandable. Minor errors are technically errors, but do not disrupt the flow or hinder comprehension. For each error span, please deduct corresponding points based on following reference points: -25="Cirtical", -5="Major", -1="Minor", -0.1="Minor/Fluency/Punctuation:0.1", 0="No-error".

---

**OUTPUT:**
`<think>`\n Okay, let's tackle this translation quality assessment task. First, I need to analyze the translation and classify the errors.
Critical: {`Critical Error ESA`}
Major: {`Major Error ESA`}
Minor: {`Minor Error ESA`}
Finally, I can calculate the final score: {`Score Calculation Process`}
`<\think>`

Score: {`Final Score`}

