# OpenReview forum: "Are Large Reasoning Models Good Translation Evaluators? Analysis and Performance Boost"
_NeurIPS.cc/2025/Conference — NeurIPS 2025 poster_

### Official Review · Reviewer_RoLH · 2025-06-16

**Clarity:** 2
**Significance:** 3
**Originality:** 3
**Rating:** 4
**Confidence:** 4

**Summary:**

This paper investigates the capability of Large Reasoning Models (LRMs) to serve as evaluators for machine translation (MT) quality, a relatively unexplored topic. The authors identify key challenges in using LRMs for MT evaluation, such as inefficient “overthinking,” scoring overestimation, and material sensitivity. They propose a solution named ThinMQM—a thinking-calibrated MQM scoring method—that aligns LRM reasoning trajectories with human-like annotations using synthetic data. The method significantly improves correlation with human MQM ratings while reducing computation by 35x across multiple LRM scales. Empirical results on WMT24 metrics benchmarks support their claims, showing strong improvements over both traditional metrics (BLEU, COMET) and LLM-based baselines (GEMBA-MQM).

**Questions:**

1.The proposed ThinMQM is evaluated only under the MQM framework and WMT24 language pairs. Could the authors clarify whether the method can generalize to other translation evaluation paradigms (e.g., adequacy-only scales) or low-resource settings? Demonstrating broader applicability would increase the impact of this work.
2.In setups where auxiliary models (e.g., Qwen-32B) are used to score, how do the authors disentangle the effect of the LRM's reasoning from the scoring model? It remains unclear whether the observed improvement is due to better annotations or just better post-processing. More controlled ablation here would strengthen the claims.
3.Can the authors provide examples or an error taxonomy where ThinMQM still diverges significantly from human judgments? This would help clarify whether current improvements come at the cost of under- or over-penalizing certain error types, and guide future work in more targeted calibration.

**Ethical Concerns:**

["NO or VERY MINOR ethics concerns only"]

**Limitations:**

Yes

**Quality:**

2

**Strengths And Weaknesses:**

Stengths:
1.This is the first systematic study of LRM-as-a-judge for MT evaluation, bridging reasoning models and MT metrics—a timely and significant contribution.
2.ThinMQM is a conceptually elegant and practically effective method to calibrate LRMs using synthetic reasoning-aligned supervision. This allows efficiency gains without sacrificing (and even improving) performance.
3.The paper thoroughly analyzes failure modes, such as overestimation and overthinking, and leverages Shapley value analysis to isolate input contributions—rare in MT metric research.
4.The paper is clearly written and well-structured, and includes useful figures/tables to illustrate thinking budgets, score distributions, and model comparisons.
Weakness:
1.The main training and evaluation are based on just three language pairs, and training data is synthesized from WMT23 MQM annotations. Generalizability to low-resource or morphologically rich languages is unclear.
2.While ThinMQM reduces variance and improves scoring alignment, the diversity of human annotation styles or reasoning is not captured—raising questions about overfitting to one “style” of evaluation.
3.When auxiliary models are used for scoring (e.g., Qwen-32B), attribution becomes blurry. It is unclear whether improvements stem from the LRM itself or the scorer, despite some efforts to address this.
4.Although the paper shows efficiency improvements, some segment-level performance under certain decoding parameters still fluctuates slightly, suggesting calibration is sensitive to model internals.

---

> ### Author Rebuttal · Authors · 2025-07-31
>
> Thank you for your constructive feedback. We appreciate the opportunity to clarify our contributions and address the concerns raised. Below are our point-by-point responses.
>
> > **Weakness 1. & Q1.** The main training and evaluation are based on just three language pairs, and training data is synthesized from WMT23 MQM annotations. Generalizability to low-resource or morphologically rich languages is unclear.
>
> We appreciate your concern regarding generalizability. The three language pairs we used come are all available MQM evaluation materials in the WMT24 benchmarks. To better assess the robustness of ThinMQM, we have conducted two new experiments on more diverse and out-of-distribution data.
>
> a. **Evaluation on WMT22 English-Russian:** We evaluated our model on the WMT22 English-Russian language pair. Although this dataset predates the knowledge cutoff for the LRM, which could be a potential source of data leakage, the results in the table below show that ThinMQM still achieves a significant improvement.
>
> *Note: ( $\Delta$ = ThinMQM - Original LRM)*
>
> |                      |  $\Delta$  ThinMQM 7B |  $\Delta$  ThinMQM 8B |  $\Delta$  ThinMQM 32B |
> | :------------------: | :----------: | :----------: | :-----------: |
> | System-Level $\rho$  |    +17.1     |     +6.6     |     +7.6      |
> | Segment-Level $\tau$ |    +19.9     |    +14.6     |     +4.5      |
>
>
> b. **A Recent Low-Resource MQM Dataset (Hindi-Chinese):** To perform a more stringent test on a low-resource language pair, we sourced a recently released Hindi-Chinese dataset with MQM annotations [1], which was published after our LRMs’ knowledge cutoff date. Since this dataset contains translations from fewer than four systems, we report segment-level Pearson and Kendall correlation. As shown in the table below, ThinMQM demonstrates strong generalization capabilities, outperforming the XCOMET-XXL baseline.
>
> | Model       | Segment-Level Pearson $\rho$ | Segment-Level $\tau$ |
> | :---------- | :--------------------------: | :------------------: |
> | XCOMET-XXL  |             62.5             |         47.8         |
> | ThinMQM 32B |           **63.4**           |       **57.4**       |
> | ThinMQM 7B  |             51.3             |         49.1         |
> | ThinMQM 8B  |             50.3             |         47.5         |
>
>
> ------
>
> > **Weakness: 2:** While ThinMQM reduces variance and improves scoring alignment, the diversity of human annotation styles or reasoning is not captured—raising questions about overfitting to one “style” of evaluation.
>
> We believe this point touches upon a fundamental challenge in evaluation research: the trade-off between aligning with a specific benchmark to achieve high correlation and capturing the full diversity of human judgment. A deep exploration of annotation "style" is a valuable research direction that extends beyond the immediate scope of this paper.
>
> Our work is situated within the MQM framework, which itself imposes a structured and detailed set of guidelines for human annotators. This inherent structure helps standardize annotations and reduce, though not eliminate, the impact of individual preferences.
>
> We will add a discussion of this point in the revised manuscript. As a potential avenue for future work, one could collect data from annotators with diverse backgrounds using varied evaluation criteria to explicitly model and understand these stylistic differences.
>
> > **Weakness 3. & Q2.** When auxiliary models are used for scoring (e.g., Qwen-32B), attribution becomes blurry. It is unclear whether improvements stem from the LRM itself or the scorer, despite some efforts to address this.
>
> We would like to address two (possible) misunderstandings:
>
> 1. The significance test in Section 3.1 compared LRM+Auxiliary Model versus the Auxiliary Model alone, showing no significant difference between these approaches. However, the LRM+Auxiliary Model outperforms the LRM alone. This indicates that the performance improvements primarily come from the Auxiliary Model rather than the LRM itself.
> 2. **ThinMQM does not use an auxiliary scorer.** We apologize if our writing was unclear. The final score in ThinMQM is generated directly and solely by the LRM as the final step in its internal reasoning trace.
>
> **Besides, for ThinMQM, the error annotation process is integral to performance.** We conducted a new ablation study. We trained the LRM to **only** predict the final score, without the intermediate error annotation steps. The results below show a drop in performance, confirming that the fine-grained error analysis is critical to ThinMQM's success.
>
> *Note: The 'w/o ESA' row indicates the performance change ($\Delta$) compared to the original ThinMQM model.*
>
> |              | Avg. (All) | En-De (Sys) | En-De (Seg) | En-Es (Sys) | En-Es (Seg) | Ja-Zh (Sys) | Ja-Zh (Seg) |
> | -----------: | :--------: | :---------: | :---------: | :---------: | :---------: | :---------: | :---------: |
> |  ThinMQM 7B |            |             |             |             |             |             |             |
> |      $\Delta$ w/o ESA |   -0.80    |    +1.00    |    -2.40    |    -0.30    |    0.00     |    -0.90    |    -2.20    |
> |  ThinMQM 8B |            |             |             |             |             |             |             |
> |      $\Delta$ w/o ESA |   -0.60    |    +0.20    |    -1.10    |    -1.30    |    0.00     |    -1.70    |    -0.20    |
> | ThinMQM 32B |            |             |             |             |             |             |             |
> |     $\Delta$ w/o ESA |   -0.70    |    -1.80    |    -1.60    |    +1.70    |    -0.90    |    -0.50    |    -0.70    |
>
>
> > **Weakness 4:** Although the paper shows efficiency improvements, some segment-level performance under certain decoding parameters still fluctuates slightly, suggesting calibration is sensitive to model internals.
>
> Thank you for noticing this. In line with common practices for LRMs [2-3], we recommend using a decoding temperature of 0.6 in practice, which we have also found provides more stable results for ThinMQM models.
>
> To formally verify the stability of this setting, we performed 3  runs at a fixed temperature of 0.6. The table below reports the mean and standard deviation. The low standard deviation indicates that performance is highly consistent and not subject to significant random fluctuations.
>
> *Note: (Mean ± Std. of 3 runs)*
>
> | **Model**   | **Avg. (All)** | **En-De (Sys)** | **En-De (Seg)** | **En-Es (Sys)** | **En-Es (Seg)** | **Ja-Zh (Sys)** | **Ja-Zh (Seg)** |
> | :---------- | :------------: | :-------------: | :-------------: | :-------------: | :-------------: | :-------------: | :-------------: |
> | ThinMQM 32B |  72.0 ± 0.003  |  80.7 ± 0.026   |  53.1 ± 0.006   |  80.8 ± 0.002   |  68.9 ± 0.002   |  92.5 ± 0.011   |  55.9 ± 0.002   |
> | ThinMQM 8B  |  70.4 ± 0.004  |  83.1 ± 0.021   |  48.6 ± 0.003   |  81.3 ± 0.020   |  68.2 ± 0.001   |  90.3 ± 0.013  |  51.0 ± 0.001   |
> | ThinMQM 7B  |  70.0 ± 0.002  |  85.4 ± 0.011   |  48.2 ± 0.004   |  77.6 ± 0.002   |  68.1 ± 0.001   |  89.3 ± 0.004  |  51.4 ± 0.003   |
>
> ------
>
> > **Q3:** Can the authors provide examples or an error taxonomy where ThinMQM still diverges significantly from human judgments? This would help clarify whether current improvements come at the cost of under- or over-penalizing certain error types, and guide future work in more targeted calibration.
>
> This is an excellent suggestion. We agree that an error analysis provides deeper insight into LRMs’ behavior. We have conducted a detailed analysis of the cases where scores from ThinMQM-32B diverge from human judgments. We categorized these discrepancies based on the MQM translation error taxonomy (Critical, Major, Minor).
>
> The distribution of these error types is presented in the table below. This analysis reveals that the model predictions most often misalign with human judgments on **Minor-level errors.** Furthermore, when analyzing all minor error types, we found that **accuracy/mistranslation** accounts for the highest proportion of Minor-level prediction errors. These results helps pinpoint specific areas for targeted improvements in future work.
>
> |       | Critical (%) | Major (%) | Minor (%) |
> | :---: | :----------: | :-------: | :-------: |
> | En-De |    0.05     |  12.35   |  87.60  |
> | En-Es |    0.21     |  10.29   |  89.50   |
> | Ja-Zh |    0.02     |  17.55   |  82.43   |
>
> ------
>
> **References**
>
> [1]  Yan, J., Yan, P., Chen, Y., Li, J., Zhu, X., & Zhang, Y. (2024). Benchmarking gpt-4 against human translators: A comprehensive evaluation across languages, domains, and expertise levels. *arXiv preprint arXiv:2411.13775*.
>
> [2] Guo, D., Yang, D., Zhang, H., Song, J., Zhang, R., Xu, R., ... & He, Y. (2025). Deepseek-r1: Incentivizing reasoning capability in llms via reinforcement learning. *arXiv preprint arXiv:2501.12948*.
>
> [3] Abdin, M., Agarwal, S., Awadallah, A., Balachandran, V., Behl, H., Chen, L., ... & Zheng, G. (2025). Phi-4-reasoning technical report. *arXiv preprint arXiv:2504.21318*.

---

> ### Author Response · Authors · 2025-08-07
>
> Dear Reviewer,
>
>
> Thank you once again for your thoughtful and constructive feedback. As the author-reviewer discussion deadline approaches, we would like to gently follow up to inquire whether you have had a chance to review our responses to your comments. We sincerely hope that we have addressed your concerns adequately and would be most grateful for any further thoughts you may wish to share.
>
> Should there be any remaining questions or points requiring clarification, we would be more than happy to provide additional explanation.
>
> Thank you very much for your time and kind consideration.
>
>
>
> Sincerely,
>
> Authors

---

### Official Review · Reviewer_JKag · 2025-06-28

**Clarity:** 1
**Significance:** 2
**Originality:** 2
**Rating:** 3
**Confidence:** 4

**Summary:**

This paper first shows the potential of existing large reasoning models (LRMs) in the translation quality evaluation task.
Then, on the basis of the observations, it proposes to use synthetic data for teaching human evaluation process, i.e., identifying the error spans and their severity,
followed by aggregating them into a single numeric score.
Through an experiment, the authors confirmed that the proposed method improves the evaluation accuracy,
in terms of system-level and segment-level pairwise accuracies, of LRMs, which can be sometime comparable or better to existing model-based metrics, such as COMET-22 and XCOMET.

**Questions:**

- Could you clearly define what deserves as a large reasoning model (LRM)?  The explanation in ll.28-29 "explicitly powered by a thinking-guided generation process" is too vague.

- Please clearly and properly explain the elements in mathematical formulations those pointed out as "Weaknesses."

- What does "All" in Figure 3 and Tables 1 and 2 mean?  Are SPE and $ACC_{eq}^{*}$ additive, while they are based on different measurement and thus have different distribution?

- Tables 1 and 2 shows the results with significance testing with $p<0.05$; which configuration is compared with each one?

- One of the conclusions in this paper would be "our thinMQM models outperform both existing LLM-based and metric-based methods" in ll.277-278, but this is not always true.
  How did the authors justify this?

Followings are comments for making charts appropriate.

- Figure 1 should also show LLM-based and metric-based methods.  While using log-scale itself is OK, using only irregular scale marks make it difficult to understand it.

- Figure 2 is first mentioned in page 8, l.256, so it should be moved to page 8, or it should be clearly explained earlier since presents the proposed method.

- Figure 3 should be located after Table 1, following the order of their first mentions in the running text.

- Figure 4: scale for p-value is strange; it is neither linear nor log-scaled.

- Figure 6 is never mentioned in the running text.

- Figure 7: line charts are inappropriate, since the horizontal axis is not interval scale but nominal scale.

- Figure 8: bar charts are not based on 0 but arbitrary values, incorrectly exaggerating the difference with height and area of bars.

References have a number of errors, including inappropriate lowercasing and inconsistent sets of BiBTeX fields.

**Ethical Concerns:**

["NO or VERY MINOR ethics concerns only"]

**Final Justification:**

Through the rebuttal, some of my misunderstandings were resolved.
Yet, I think this paper should be judged once it is revised.
For fairness, I've not taken care of new results provided after the submission deadline.

**Limitations:**

Yes.

**Paper Formatting Concerns:**

Nothing.

**Quality:**

2

**Strengths And Weaknesses:**

Strengths:

- The paper unveils several tendencies of existing LRMs in the translation quality evaluation task (Section 3).
- It proposes an effective way to improve the accuracy of the evaluation metric based on LRM, relying on synthetic data that follow manual evaluation process (Section 4).

Weaknesses:

- While the analysis of tendencies of simple use of LRMs is informative, it is too detailed, so as to make the main part not self-contained.

- While the proposed method improves evaluation accuracy of LRMs,
  they are still worse than existing smaller task-specific models, e.g., XCOMET (3B or 10B).

- Mathematical formulations are imprecise and/or defective.  For example, $S'$ in l.154 is never used;
  the operation $+$ for two textual elements in l.157 is undefined; $\phi_{s}^{approx}$ and $\phi{r}^{approx}$ are not defined and their usages are not explained.
  while $T_{ESA}$ and $T_{score}$ are defined as function in ll.252-253, they are used as variables in l.255 and Equation (3);
  the loss function in Equation (3), $\mathcal{L}$, is never defined; the returned value of $M(\cdot;\theta)$ is unknown.

---

> ### Author Rebuttal · Authors · 2025-07-31
>
> Thank you for your time and detailed feedback, which is invaluable for improving our work. We would like to address the concerns raised and clarify some points.
>
> ------
>
> > **Weakness 1:** While the analysis of tendencies of simple use of LRMs is informative, it is too detailed, so as to make the main part not self-contained; the method of generating the proposed synthetic data is not explained in the main part at all.
>
> Thank you for acknowledging that our analysis is informative. We would like to gently clarify that our research is *analysis-driven*. The detailed analysis of potential issues in current LRM practices in MT evaluation is fundamental to motivating and deriving our final proposed method. Therefore, we consider this analytical section to be a crucial component of the paper's narrative.
>
> However, we understand the concern regarding the balance of the paper. We are more than willing to revise the proportions and level of detail of the analysis and method sections to improve the overall flow and focus.
>
> Regarding the method for generating synthetic data, we would like to clarify that this was included in the main paper. We provide a detailed **description in Lines 246-266**. We apologize if its placement was not prominent enough and will ensure it is clearly signposted in the revised version.
>
> ------
>
> > **Weakness 2:** While the proposed method improves evaluation accuracy of LRMs, they are still worse than existing smaller task-specific models, e.g., XCOMET (3B or 10B).
>
> We would like to clarify two key points:
>
> 1. The primary contribution of our paper is the improvement of LRMs-as-Judges. Our goal is to analyze specific problems and enhance this paradigm.
> 2. The comparison with XCOMET may also consider the **difference in training data**. Our method uses only ~12k synthetic trajectories (for 2 language pairs), whereas XCOMET was trained on 1M Direct Assessment (DA) human annotations (36 languages) and 176K MQM annotations (14 languages) [1].
>
> Furthermore, as we noted in our response to other reviewers, a key strength of the LRM-based approach lies in its potential for out-of-distribution and low-resource language pairs. This is demonstrated by our additional experiments on recently released Hindi-Chinese MQM data [2].
>
>
> | Model       | Segment-Level Pearson $\rho$ | Segment-Level $\tau$ |
> | :---------- | :--------------------------: | :------------------: |
> | XCOMET-XXL  |             62.5             |         47.8         |
> | ThinMQM 32B |           **63.4**           |       **57.4**       |
> | ThinMQM 7B  |             51.3             |         49.1         |
> | ThinMQM 8B  |             50.3             |         47.5         |
>
>
> ------
>
> > **Weakness 3:** Mathematical formulations are imprecise and/or defective.
>
> Thank you very much for your detailed feedback regarding the mathematical formulations. We noted that some of these concepts were defined within the paper, and we aim to clarify any misunderstandings below. In the revised manuscript, we will incorporate more detailed explanations for these notions and refine our notation to ensure greater clarity and precision.
>
> - **Implementation of $v(\cdot)$ in l.153:** As we explained in Lines 153-154, $v(\cdot)$ is calculated "using system-level and segment-level metrics," which correspond to the SPA and $Acc{_{Eq}^*}$ metrics defined in Line 133. We will make this link more explicit.
> - **$S’$ in l.154 is never used:** This is a typo and should be the lowercase $s'$ from Equation 2. We will correct this.
> - **The $+$ operation in l.157:** The notation $r+t$ was intended to represent the standard reference-based evaluation setting (Ref.). Thus, $v(r+t)$ is the correlation score under this setting. If the $+$ symbol is confusing, we are happy to replace it with a more descriptive notation (e.g., $v([r, t])$) and define it clearly.
> - **$\phi_{s}^{approx}$ and $\phi_{r}^{approx}$ are not defined:** We used "approximate" here to distinguish from the strict definition of Shapley Value in Eq. 2. As noted in Line 154-157, a strict calculation would require evaluating all subsets. However, in the context of machine translation evaluation, evaluating a reference $v(r)$ or a translation $v(t)$ in isolation is not a valid setting. Our $\phi_{s}^{approx}$ is the practical Shapley Value for the MT evaluation task. We will clarify this and can rename it to $\phi_{s}^{MT}$  for better understanding.
> - **$T_{ESA}$  and $T_{score}$ usage in l.255 and Eq.3:** Our intention was to represent the *output* of these functions after the mapping. We will revise the notation to be more explicit, for instance: $[T_{ESA}(X), T_{Score}(T_{ESA}(X))]$.
> - **The loss function $\mathcal{L}$ in Eq.3 is never defined:** We used the cross-entropy loss function. We will explicitly state this in the experimental setup (Section 4.2) in the revision.
> - **The returned value of $M(·;θ)$ is unknown:** As defined in **Line 257**, $M$ is the model. The return value $M(·;θ)$ is the **generated output sequence** from the model. We will ensure this is clear.
>
> ------
>
> > **Q1:** Could you clearly define what deserves as a large reasoning model (LRM)? The explanation in ll.28-29 "explicitly powered by a thinking-guided generation process" is too vague.
>
> Thank you for this question. We appreciate the opportunity to clarify the definition of Large Reasoning Models (LRMs) with reference to existing literature:
>
> - *A Large Reasoning Models (LRMs) is an advanced artificial intelligence model that extends the capabilities of traditional Large Language Models (LLMs) by introducing the concept of a "thought". A "thought" is defined as a sequence of tokens representing intermediate reasoning steps, designed to mimic complex human cognitive processes like tree search and reflective thinking [3].* Representative LRMs include OpenAI's o1/o3, Google’s Gemini 2.5 Pro, DeepSeek-R1, etc.
>
> We will summarize and refine this definition in the paper to be more concrete.
>
> ------
>
> > **Q2:** What does "All" in Figure 3 and Tables 1 and 2 mean? Are SPA and $Acc{_{Eq}^*}$ additive, while they are based on different measurement and thus have different distribution?
>
> "All" refers to the **mean (average) score** of all correlation metrics used. We follow the official evaluation practice of the WMT24 MQM shared task [4], which also reports the average of different correlation coefficients to provide a measure of performance.
>
> ------
>
> > **Q3:** Tables 1 and 2 shows the results with significance testing with p<0.05; which configuration is compared with each one?
>
> As described in **Lines 139-140**, the testing was conducted using the permutation-based method (1,000 resampling times), i.e., PERM-BOTH hypothesis test [5]. This also follows WMT24 official evaluation practice [4].
>
> ------
>
> > **Q4:** One of the conclusions in this paper would be "our thinMQM models outperform both existing LLM-based and metric-based methods" in ll.277-278, but this is not always true. How did the authors justify this?
>
> We will revise it to be more precise. The corrected statement will clarify that our ThinMQM-32B model achieves superior *average* performance compared to the baselines, though not necessarily on every individual language pair. We will also refine our conclusion to emphasize that our method shows a highly competitive performance and a much better efficiency-performance trade-off, particularly within the scope of LRM/LLM-based evaluators.
>
> ------
>
> > **Response to Comments on Charts**
>
> - **Figure 1:** For part (a), we will include the available models in the scale comparison. However, some closed-source models, such as GEMBA-ESA, lack publicly available information regarding their scale. For (b), we would like to clarify that the scope is limited to LRMs as we are analyzing reasoning budgets.
> - **Figure 7:** We will add an axis break on the horizontal axis to improve readability.
> - **General:** We will review all figures to improve their clarity and adjust their placement within the paper based on the feedback.
>
> ------
>
> **References**
>
> [1] Guerreiro, N. M., Rei, R., Stigt, D. V., Coheur, L., Colombo, P., & Martins, A. F. (2024). XCOMET: Transparent machine translation evaluation through fine-grained error detection. *Transactions of the Association for Computational Linguistics*, *12*, 979-995.
>
> [2] Yan, J., Yan, P., Chen, Y., Li, J., Zhu, X., & Zhang, Y. (2024). Benchmarking gpt-4 against human translators: A comprehensive evaluation across languages, domains, and expertise levels. *arXiv preprint arXiv:2411.13775*.
>
> [3] Xu, F., Hao, Q., Zong, Z., Wang, J., Zhang, Y., Wang, J., ... & Li, Y. (2025). Towards large reasoning models: A survey of reinforced reasoning with large language models. *arXiv preprint arXiv:2501.09686*.
>
> [4] Freitag, M., Mathur, N., Deutsch, D., Lo, C. K., Avramidis, E., Rei, R., ... & Lavie, A. (2024). Are LLMs breaking MT metrics? results of the WMT24 metrics shared task. In *Proceedings of the Ninth Conference on Machine Translation* (pp. 47-81).
>
> [5] Deutsch, D., Dror, R., & Roth, D. (2021). A statistical analysis of summarization evaluation metrics using resampling methods. *Transactions of the Association for Computational Linguistics*, *9*, 1132-1146.

---

> ### Comment · Reviewer_JKag · 2025-08-06
>
> Thanks for your rebuttal.
>
> I understand that some contents in my review, such as description about synthetic data and implementation of $v(\cdot)$, were not correct, and will take down them accordingly.
> Apologies for the former; I've put many notes there but somehow forgot them when writing the review.
>
> Let me elaborate on some of my previous questions.
>
> >> Q3: Tables 1 and 2 shows the results with significance testing with p<0.05; which configuration is compared with each one?
>
> I'm asking the system compared with each tested system to evaluate its significance.
>
> > Figure 1: For part (a), we will include the available models in the scale comparison. However, some closed-source models, such as GEMBA-ESA, lack publicly available information regarding their scale. For (b), we would like to clarify that the scope is limited to LRMs as we are analyzing reasoning budgets.
>
> I understand only open-weight models can be included.
> I imagine xtics 5, 50, and 500 for (a) and 10, 100, and 1000 for (b) would be good enough.
>
> > Figure 7: We will add an axis break on the horizontal axis to improve readability.
>
> - If you regard these four as nominal items (e.g., apple, orange, banana, strawberry), please avoid linking their values with a line.
> - If you consider the relative size is important and thus line chart makes sense, please locate them with appropriate horizontal position.  In this case, an axis break does not solve the inappropriateness; gradient of the line over the break misleads the magnitude of the score gap.
>
> > General: We will review all figures to improve their clarity and adjust their placement within the paper based on the feedback.
>
> Glad to know you'll improve clarity.  Yet, please note that their appropriateness must be a top priority.

---

> > ### Author Response · Authors · 2025-08-06
> >
> > Thank you for engaging in the discussion and your detailed follow-up. We are glad our previous responses helped clarify some misunderstanding points, and we appreciate the opportunity to address your remaining concerns.
> >
> > ---
> >
> > > **Q3: Significance Testing.** I'm asking the system compared with each tested system to evaluate its significance.
> >
> > The methodology for the significance tests ($p$-value<0.05) is as follows:
> >
> > - **For Table 1**, which analyzes the impact of different evaluation materials (*Src.*, *Ref.*, *Joint.*), we compare the **best-performing setup against the other two setups** **within the same model block.** A dagger (†) indicates that the best result is statistically significant compared to both of the other two results. For example, for the model *Deepseek-R1* on the En-De Acc.$^*_{eq}$ metric, the score of 47.4 for the *Src.* setup is marked with a dagger because it is significantly higher ($p$-value<0.05) than the scores for both *Ref.* (43.0) and *Joint.* (46.6).
> > - **For Table 2**, which compares the performance of different models, we compare the **best-performing model against all other models for each column**. A dagger (†) indicates that this top score is significantly better than all other scores in that column. For example, for the Ja-Zh Acc.$^*_{eq}$ metric, our *ThinMQM* model's score of 56.1 is marked because it is significantly better ($p$-value<0.05) than *all other models*, including the second-best score of 53.9 from *Gemba-ESA*.
> >
> > We will revise the captions of both tables in the final version to be more explicit.
> >
> > ---
> >
> > > **Figure 1.** I understand only open-weight models can be included. I imagine xtics 5, 50, and 500 for (a) and 10, 100, and 1000 for (b) would be good enough.
> >
> > Thank you for the concrete suggestions for the axis ticks. We will update Figure 1 with the tick marks you recommended: 5, 50, 500 for part (a) and 10, 100, 1000 for part (b).
> >
> > ---
> >
> > > **Figure 7.** If you regard these four as nominal items (e.g., apple, orange, banana, strawberry), please avoid linking their values with a line. If you consider the relative size is important and thus line chart makes sense, please locate them with appropriate horizontal position.
> >
> > We appreciate your thoughtful consideration of the figure's design. To address this, we will regard these **four as nominal item** and **removing the connecting line between the data points for “#Thinking Tokens”.**
> >
> > ---
> >
> > Finally, we sincerely appreciate you emphasizing the importance of appropriate figure presentation. Your guidance on this will improve the clarity and quality of our paper. We will carefully revise all figures and the paper based on your suggestions.

---

### Official Review · Reviewer_EpwK · 2025-07-04

**Clarity:** 4
**Significance:** 2
**Originality:** 3
**Rating:** 5
**Confidence:** 3

**Summary:**

The paper delivers the first comprehensive examination of large reasoning models (LRMs) as machine-translation judges, showing that while their “slow-thinking” paradigm can model the MQM evaluation process, they also require carefully chosen inputs, often “over-think” easy segments and systematically over-estimate quality. Building on these insights, the authors introduce ThinMQM—a post-training scheme that feeds LRMs synthetic, human-aligned error-annotation + scoring trajectories—which realigns the model’s reasoning, cuts inference-time “thinking” tokens by about 35 × and boosts human-correlation on the WMT-24 Metrics benchmark by up to +8.7 points across 7 B–32 B models, matching or surpassing state-of-the-art neural metrics without auxiliary scorers .

**Questions:**

The paper attributes the +3.9 – +8.7 point gains to ThinMQM but does not disentangle which design choices—trajectory length, explicit ESA tokens, or post-training itself—drive the lift. (More ablation studies would be nice.)
The efficiency story (35 × fewer “thinking” tokens) is promising yet lacks a cost–quality curve.

**Ethical Concerns:**

["NO or VERY MINOR ethics concerns only"]

**Final Justification:**

My concerns are well addressed and would be happy to raise my score slightly. Thank you for the experiments

**Limitations:**

yes

**Quality:**

2

**Strengths And Weaknesses:**

The paper has solid empirical design because it study evaluates four LRMs ranging from 7 B to 671 B parameters under a unified GEMBA-MQM protocol, reporting both system- and segment-level SPA/Acc with 1 000-sample permutation tests, which lends statistical rigor. Substantial gains and efficiency - ThinMQM lifts correlation by up to +8.7 points while slashing the “slow-thinking” budget ~35 × and cutting wall-clock inference from 12 min to 40 s per 1 k sentences. The writing is easy to follow-the introduction explicitly lists three research questions, and Figure 2 walks the reader through the evaluation pipeline and ThinMQM calibration steps, making the workflow flows well. First systematic look at “LRM-as-a-judge.” The paper is, one of the earliest comprehensive study of reasoning-tuned models for MQM evaluation, and it introduces ThinMQM—synthetic, human-aligned reasoning trajectories to calibrate scoring.

ThinMQM is tuned on only ~11.9 k synthetic instances covering two language pairs, raising questions about robustness beyond WMT24’s three test pairs. There are some clarity issues - there is no explanation in fig.2's caption. Moreover, the novelty in terms of using LRMs for judging the quality of a response is questionable, because there are already many papers on this. While the thin-trajectory idea is neat, the fine-tune-on-synthetic-CoT recipe resembles recent chain-of-thought distillation work; novelty lies more in application domain than in method.

---

> ### Author Rebuttal · Authors · 2025-07-31
>
> Thank you for your valuable and constructive feedback. We appreciate your insights and believe they will strengthen our paper. Below are our responses.
>
> ------
>
> > **Weakness 1:** ThinMQM is tuned on only ~11.9 k synthetic instances covering two language pairs, raising questions about robustness beyond WMT24’s three test pairs.
>
> We acknowledge the concern. The fact is that high-quality, human-annotated MQM test data is scarce due to the intensive labor and cost involved. In our work, we have utilized all publicly *available test data (3 languages)* from the WMT24 MQM shared task.
>
> To assess the generalization capabilities of ThinMQM, we have conducted two additional experiments:
>
> a. **Evaluation on WMT22 En-Ru:** We tested ThinMQM models and corresponding unaligned LRMs on the WMT22 English-Russian MQM data. This pair may help (Note: The WMT22 data was released before our LRMs' knowledge cutoff date) demonstrate the robustness of the method across different data distributions within the MQM scope. Experimental results in the following table show a consistent performance improvement.
>
> *Note: ( $\Delta$ = ThinMQM - Original LRM)*
>
> |                      |  $\Delta$  ThinMQM 7B |  $\Delta$  ThinMQM 8B |  $\Delta$  ThinMQM 32B |
> | :------------------: | :----------: | :----------: | :-----------: |
> | System-Level $\rho$  |    +17.1     |     +6.6     |     +7.6      |
> | Segment-Level $\tau$ |    +19.9     |    +14.6     |     +4.5      |
>
>
> b. **Evaluation on a Low-Resource, Out-of-Distribution Dataset:** To perform a stricter test, we sourced a recently released Hindi-Chinese dataset with MQM annotations [1], which was published after LRMs' knowledge cutoff date. Since this dataset contains translations from fewer than four systems, we report segment-level Pearson and Kendall correlation since the amount of candidate system is quite small. As shown in the table below, ThinMQM demonstrates strong correlation, performing competitively (32B model) against the powerful XCOMET-XXL baseline.
>
> | Model       | Segment-Level Pearson $\rho$ | Segment-Level $\tau$ |
> | :---------- | :--------------------------: | :------------------: |
> | XCOMET-XXL  |             62.5             |         47.8         |
> | ThinMQM 32B |           **63.4**           |       **57.4**       |
> | ThinMQM 7B  |             51.3             |         49.1         |
> | ThinMQM 8B  |             50.3             |         47.5         |
>
> These new results suggest that ThinMQM generalizes well to unseen (w.r.t. SFT data) and low-resource language pairs.
>
> ------
>
> > **Weakness 2:** There are some clarity issues - there is no explanation in fig.2's caption.
>
> Thank you for pointing this out. We agree that the caption for Figure 2 was too brief. We will revise it in the final version to provide a comprehensive explanation of our research framework and clearly define the notations used in the synthetic data generation process.
>
> ------
>
> > **Weakness 3:** Moreover, the novelty in terms of using LRMs for judging the quality of a response is questionable, because there are already many papers on this. While the thin-trajectory idea is neat, the fine-tune-on-synthetic-CoT recipe resembles recent chain-of-thought distillation work; novelty lies more in application domain than in method.
>
> We appreciate you recognizing our work's contribution and agree with your assessment that a key part of our novelty lies in the application domain.
>
> While the idea of using LRMs for quality evaluation exists, there are still significant open challenges in machine translation evaluation regarding *how* to do so effectively and efficiently. Our work addresses a piece of this puzzle by exploring tailored materials for different model capabilities, better scoring mechanisms, reducing reasoning costs while improving performance.
>
> Furthermore, we would like to respectfully clarify a distinction between our "thin-trajectory" and Chain-of-Thought distillation. Our trajectories are **not distilled** from the outputs of a larger teacher model [2,3]. Instead, they are **synthetically constructed** using templates that are explicitly designed to mimic the standard operating procedures and annotation guidelines followed by humans. This approach builds a reasoning trajectory from first principles based on ground-truth labels, rather than learning to imitate another model's reasoning.
>
> ------
>
> > **Q1:** The paper attributes the +3.9 – +8.7 point gains to ThinMQM but does not disentangle which design choices—trajectory length, explicit ESA tokens, or post-training itself—drive the lift. (More ablation studies would be nice.) The efficiency story (35 × fewer “thinking” tokens) is promising yet lacks a cost–quality curve.
>
> This is an excellent suggestion. To better disentangle the contributions of our design choices, we have conducted an ablation study on the Error Span Annotation (ESA) component.
>
> We ran an experiment where we removed the ESA spans from the reasoning trajectory during inference, retaining only the final score. The results, shown below, indicate a drop in performance. This highlights that identifying error spans is a crucial component for achieving high accuracy. Furthermore, as suggested by Reviewer RoLH, retaining ESA allows for valuable, fine-grained error analysis.
>
> *Note: The 'w/o ESA' row indicates the performance change ($\Delta$) compared to the original ThinMQM model.*
>
> |              | Avg. (All) | En-De (Sys) | En-De (Seg) | En-Es (Sys) | En-Es (Seg) | Ja-Zh (Sys) | Ja-Zh (Seg) |
> | -----------: | :--------: | :---------: | :---------: | :---------: | :---------: | :---------: | :---------: |
> |  ThinMQM 7B |            |             |             |             |             |             |             |
> |     $\Delta$ w/o ESA |   -0.80    |    +1.00    |    -2.40    |    -0.30    |    0.00     |    -0.90    |    -2.20    |
> |  ThinMQM 8B |            |             |             |             |             |             |             |
> |     $\Delta$ w/o ESA |   -0.60    |    +0.20    |    -1.10    |    -1.30    |    0.00     |    -1.70    |    -0.20    |
> | ThinMQM 32B |            |             |             |             |             |             |             |
> |     $\Delta$ w/o ESA |   -0.70    |    -1.80    |    -1.60    |    +1.70    |    -0.90    |    -0.50    |    -0.70    |
>
> Regarding the cost-quality curve, we agree this would be a valuable addition to illustrate the efficiency of our model. We are preparing this visualization and will gladly incorporate it into the final version of the paper. We appreciate your understanding that we cannot attach new figures during the rebuttal phase.
>
> ------
>
> **References**
>
> [1] Yan, J., Yan, P., Chen, Y., Li, J., Zhu, X., & Zhang, Y. (2024). Benchmarking gpt-4 against human translators: A comprehensive evaluation across languages, domains, and expertise levels. *arXiv preprint arXiv:2411.13775*.
>
> [2] Zhuang, X., Zhu, Z., Wang, Z., Cheng, X., & Zou, Y. (2025). Unicott: A unified framework for structural chain-of-thought distillation. In *The Thirteenth International Conference on Learning Representations*.
>
> [3] Dai, C., Li, K., Zhou, W., & Hu, S. (2024). Improve Student’s Reasoning Generalizability through Cascading Decomposed CoTs Distillation. In *Proceedings of the 2024 Conference on Empirical Methods in Natural Language Processing* (pp. 15623-15643).

---

> > ### Comment · Reviewer_EpwK · 2025-08-01
> >
> > Thank you a lot for the explanation and experiments! They helped clarify my concerns a lot.
> >
> > Best to have more ablations on the template (vs distillation) comparison. Since my score was already positive, I'd maintain it.

---

> > > ### Author Response · Authors · 2025-08-06
> > >
> > > Thank you very much for your timely engagement in the discussion and for providing further feedback.
> > >
> > > We have taken some time to conduct additional experiments to supplement the results regarding the *comparison between different baseline templates and our proposed ThinMQM, as you suggested*. To perform a thorough ablation on this point, we used GPT-4o to paraphrase our original prompt, creating three additional prompts, which we denote as `P1`, `P2`, and `P3`.
> > >
> > > We have integrated the results from these new prompting templates with the existing results from Table 2 in our paper and present the new comparison in the table below.
> > >
> > > ---
> > >
> > >
> > >
> > > |               Model | Avg. (All) | En-De (Sys) | En-De (Seg) | En-Es (Sys) | En-Es (Seg) | Ja-Zh (Sys) | Ja-Zh (Seg) |
> > > | ------------------: | :--------: | :---------: | :---------: | :---------: | :---------: | :---------: | :---------: |
> > > |         ThinMQM 32B |  **72.2**  |    83.2     |    52.5     |    80.7     |    69.2     |    91.3     |    56.1     |
> > > |             QwQ 32B |    68.3    |    79.8     |    46.8     |    76.1     |    68.0     |    91.9     |    46.9     |
> > > |             *w/ P1* |    68.3    |    77.8     |    47.4     |    79.3     |    68.0     |    89.8     |    47.4     |
> > > |             *w/ P2* |    68.1    |    77.0     |    46.9     |    74.4     |    68.0     |    93.7     |    48.4     |
> > > |             *w/ P3* |    68.3    |    79.3     |    46.9     |    78.2     |    68.0     |    90.9     |    46.4     |
> > > |                     |            |             |             |             |             |             |             |
> > > |          ThinMQM 8B |  **70.8**  |    85.5     |    48.6     |    81.3     |    68.2     |    90.5     |    51.0     |
> > > | R1-Distill-Llama-8B |    64.9    |    71.8     |    42.9     |    78.5     |    68.0     |    84.7     |    43.5     |
> > > |             *w/ P1* |    64.5    |    74.5     |    42.9     |    72.3     |    68.0     |    85.7     |    43.5     |
> > > |             *w/ P2* |    62.6    |    71.1     |    42.9     |    65.5     |    68.0     |    84.3     |    43.5     |
> > > |             *w/ P3* |    63.4    |    70.2     |    42.9     |    72.2     |    68.0     |    83.3     |    43.5     |
> > > |                     |            |             |             |             |             |             |             |
> > > |          ThinMQM 7B |  **69.8**  |    84.5     |    48.5     |    77.8     |    68.0     |    89.0     |    51.3     |
> > > |  R1-Distill-Qwen-7B |    61.1    |    67.3     |    42.9     |    61.0     |    68.0     |    83.8     |    43.5     |
> > > |             *w/ P1* |    62.5    |    70.8     |    42.9     |    70.0     |    68.0     |    79.6     |    43.5     |
> > > |             *w/ P2* |    59.7    |    63.1     |    42.9     |    58.5     |    68.0     |    82.4     |    43.5     |
> > > |             *w/ P3* |    62.1    |    66.5     |    42.9     |    69.6     |    68.0     |    81.9     |    43.5     |
> > >
> > > ---
> > >
> > > As the results indicate, **ThinMQM still demonstrates a advantage across a range of model sizes**. Furthermore, these additional experiments provide some valuable takeaways:
> > >
> > > - The performance of larger models (32B) is not significantly impacted by variations in the prompt templates.
> > > - Smaller models (7B, 8B) exhibit greater sensitivity to changes in the prompt; however, the resulting performance fluctuations are limited and still do not surpass the results achieved by ThinMQM.
> > >
> > > We hope these additional ablations address the concerns you raised during the rebuttal period. We appreciate your constructive feedback and will include these results in the revised paper.

---

### Official Review · Reviewer_edhE · 2025-07-10

**Clarity:** 4
**Significance:** 2
**Originality:** 2
**Rating:** 6
**Confidence:** 3

**Summary:**

This paper is the first systematic evaluation of large reasoning models (LRMs) as the judge for evaluating machine translation tasks. The authors systematically analyze LRM performance in MT evaluation and identify key insights:

1) Smaller LRM models (7B–8B) benefit more from reference-based inputs, while larger ones (32B–671B) perform better with source-based information. This contradicts prior LLM findings and suggests that LRM design and scale fundamentally alter how evaluation context is processed.
2) Although reference-based evaluation is popular, it may hurt LRM accuracy, especially at scale, due to noise or low quality in reference translations. Human MQM annotations are reference-free, so introducing references may create a mismatch with human judgment standards.
3) Switching from rule-based to model-based re-scoring (using auxiliary models like Qwen-2.5 32B) increases computation but doesn't significantly improve alignment with human evaluations. Worse, the improvements often stem from the scorer rather than the LRM, muddying attribution.
4) Larger LRMs often generate more tokens and reasoning steps (higher “thinking budget”), but this does not consistently correlate with better evaluation accuracy. Surprisingly, correct judgments require more effort only in edge cases, and LRMs still underperform general LLMs in nearly half of tested scenarios.

To address these issues, the paper proposes Thinking-calibrated MQM (ThinMQM) scoring, which involves training LRMs on synthetic, human-like thinking trajectories using supervised fine-tuning. Experiments on WMT24 Metrics benchmarks demonstrate that this approach reduces thinking budgets while improving evaluation performance across various LRM.

**Questions:**

1) How does changing the rule-based scoring schema change the correlation with human judgement? For example instead of -25, -5, and -1 if you use -3, -2, -1 or -100, -50, -1 how does that affect your evaluation and correlation with human judgement?

2) You are using a quite high temperature, 0.6 for decoding, which means more randomness in the experiments, and less reproducibility. Is there any reason that you didn’t use temperature 0 or 0.1 instead?

3) Why does your model perform better than XCOMET on Ja-Zh tasks but not on others? Do you think the reason is that XCOMET does not have the Chinese and Japanese knowledge or is there any other reason?

**Ethical Concerns:**

["NO or VERY MINOR ethics concerns only"]

**Final Justification:**

The authors responded to all my comments and have resolved my concerns

**Limitations:**

YES

**Quality:**

3

**Strengths And Weaknesses:**

Strengths:

1) This paper is the first systematic evaluation of LRM as the evaluators on the machine translation tasks, which provides a lot of interesting insights.
2) The paper is very well written and clear, make it easy to understand and follow for even less familiar to the topic researchers,
3) This paper studies each concept in a very detailed evaluation setup and with many observations, that does not leave any questions around the topic unanswered.

Weaknesses:

1) The results suggest that XCOMET, an Encoder only model that is trained with a backbone encoder of size 3.5B is performing better than the proposed trained LRM method in tasks when one side of the translation is English. Additionally, since this is an encoder only model it is cheaper than LRM models as well.
2) The paper does not study more powerful commercial LLMs such as GPT-O3 or Gemini 2.5 Pro. It would be interesting to see how they will perform in these tasks and what are the problems with those frontier LLMs.

---

> ### Author Rebuttal · Authors · 2025-07-31
>
> Thank you for your valuable feedback and insightful questions. Below are our point-by-point responses.
>
> > **Weaknesses 1:** The results suggest that XCOMET, an Encoder only model that is trained with a backbone encoder of size 3.5B is performing better than the proposed trained LRM method in tasks when one side of the translation is English. Additionally, since this is an encoder only model it is cheaper than LRM models as well.
>
> Firstly, we would like to respectfully clarify a crucial difference in the **training data** used. Our method was trained on ~**12k** synthetic reasoning trajectories. In contrast, XCOMET [1] utilizes much larger datasets: the WMT 17-20 Direct Assessment (DA) dataset with **1M samples** across 36 languages, and an additional **176k samples** from WMT20-22 and other MQM-annotated sources.
>
> Regarding research focus, our primary contribution is to analyze the challenges and potential improvements of using Large Reasoning Models (LRMs) for MT evaluation, rather than solely pursuing SOTA performance.
>
> Inspired by your comment & Q3, we conducted an additional experiment on a **low-resource language pair** to further investigate the advantage of LRM backbones. On a most recent publicly available Hindi-Chinese MQM-annotated dataset [2], our ThinMQM model outperforms XCOMET-XXL, as shown in Table below. This suggests that while our LRM approach has a larger parameter count, it may offer valuable insights into out-of-distribution generalization and better performance on low-resource language pairs.
>
> | Model       | Segment-Level Pearson $\rho$ | Segment-Level $\tau$ |
> | :---------- | :--------------------------: | :------------------: |
> | XCOMET-XXL  |             62.5             |         47.8         |
> | ThinMQM 32B |           **63.4**           |       **57.4**       |
> | ThinMQM 7B  |             51.3             |         49.1         |
> | ThinMQM 8B  |             50.3             |         47.5         |
>
>
> ------
>
> > **Weakness 2:** The paper does not study more powerful commercial LLMs such as GPT-O3 or Gemini 2.5 Pro. It would be interesting to see how they will perform in these tasks and what are the problems with those frontier LLMs.
>
> We agree that evaluating frontier models like GPT-o3 or Gemini-2.5 Pro would be very interesting. However, the APIs for these models do not offer **transparent reasoning trajectories**, which are essential for the type of fair and detailed analysis central to our study. For instance, the Gemini 2.5 Pro API only returns **synthesized** reasoning trajectories [3]. Our choice of DeepSeek-R1 was therefore motivated by its transparent reasoning trajectories, making it a more suitable model for our analysis.
>
> Moreover, MT evaluation is a token-intensive task, requiring the assessment of multiple system outputs for each source sentence. Even with the relatively affordable DeepSeek-R1 API, our experiments cost nearly $800.
>
> However, to provide a more direct comparison, we have run an additional experiment with Gemini 2.5 Pro during the rebuttal period. As shown in the table below, its performance is superior to the GPT-4-driven GEMBA but still falls short of our ThinMQM method. We will add this result to the paper in the future.
>
> | **Model**      | **Avg. (All)** | **En-De (Sys)** | **En-De (Seg)** | **En-Es (Sys)** | **En-Es (Seg)** | **Ja-Zh (Sys)** | **Ja-Zh (Seg)** |
> | :------------- | :------------: | :-------------: | :-------------: | :-------------: | :-------------: | :-------------: | :-------------: |
> | ThinMQM 32B    |    **72.2**    |      80.8       |      53.2       |      81.1       |      68.8       |      93.3       |      55.7       |
> | GEMBA-ESA      |      71.1      |      79.1       |      50.7       |      84.0       |      68.3       |      90.8       |      53.9       |
> | Gemini-2.5 Pro |      71.0      |      82.3       |      51.2       |      76.9       |      68.0       |      94.8       |      53.1       |
>
> ------
>
> > **Q1:** How does changing the rule-based scoring schema change the correlation with human judgement? For example instead of -25, -5, and -1 if you use -3, -2, -1 or -100, -50, -1 how does that affect your evaluation and correlation with human judgement?
>
> This is an insightful question. We ran an experiment to test a different scoring scheme (i.e., -3/-2/-1). In Table below, we averaged the change in correlation metrics across all languages and evaluation levels.
> We found that while changing the weights does affect the absolute correlation scores, the differences are not very substantial.
>
> *Note: $\Delta$ = New scale $-$ MQM scale*
>
> |                              | R1 671B | QwQ 32B | R1-Distill-Llama 8B | R1-Distill-Qwen 7B |
> | ---------------------------: | ------- | :-----: | :-----------------: | :----------------: |
> | Avg. $\Delta$ (Acc.$^*_{eq}$ + SPA) | +0.23 $\uparrow$   |  +0.33  $\uparrow$  |       -0.73  $\downarrow$        |       -0.50 $\downarrow$       |
>
> A possible reason is that meta-evaluation metrics are primarily sensitive to the **rank order** of the segments. As long as the ordinal relationship of the penalties is preserved (i.e., major errors receive a larger penalty than minor errors), the rankings remain relatively stable, thus ensuring the scoring's reasonability.
>
> ------
>
> > **Q2:** You are using a quite high temperature, 0.6 for decoding, which means more randomness in the experiments, and less reproducibility. Is there any reason that you didn’t use temperature 0 or 0.1 instead?
>
> Our reasons for this choice are threefold:
>
> 1. **Recommended Practice.** This temperature setting (=0.6 or >0.6) is often the recommended setting for complex reasoning tasks with LLMs and has been used in prior work [4-6].
> 2. **Our Empirical Results.** Our preliminary experiments showed that lower temperatures (e.g., 0 or 0.1) led to a noticeable degradation in performance on this task (Figure 8b, QwQ-32B). We have also included significance tests in the paper for all comparison experiments.
> 3. **Additional Multi-run Experiments.** To explicitly address the concern of randomness, we ran our ThinMQM model 3 times at temperature=0.6. The results below show a very small standard deviation, indicating that the randomness has a minimal impact on the final outcome.
>
> *Note: (Mean ± Std. of 3 runs)*
>
> | **Model**   | **Avg. (All)** | **En-De (Sys)** | **En-De (Seg)** | **En-Es (Sys)** | **En-Es (Seg)** | **Ja-Zh (Sys)** | **Ja-Zh (Seg)** |
> | :---------- | :------------: | :-------------: | :-------------: | :-------------: | :-------------: | :-------------: | :-------------: |
> | ThinMQM 32B |  72.0 ± 0.003  |  80.7 ± 0.026   |  53.1 ± 0.006   |  80.8 ± 0.002   |  68.9 ± 0.002   |  92.5 ± 0.011   |  55.9 ± 0.002   |
> | ThinMQM 8B  |  70.4 ± 0.004  |  83.1 ± 0.021   |  48.6 ± 0.003   |  81.3 ± 0.020   |  68.2 ± 0.001   |  0.903 ± 0.013  |  51.0 ± 0.001   |
> | ThinMQM 7B  |  70.0 ± 0.002  |  85.4 ± 0.011   |  48.2 ± 0.004   |  77.6 ± 0.002   |  68.1 ± 0.001   |  0.893 ± 0.004  |  51.4 ± 0.003   |
>
> ------
>
> > **Q3:** Why does your model perform better than XCOMET on Ja-Zh tasks but not on others? Do you think the reason is that XCOMET does not have the Chinese and Japanese knowledge or is there any other reason?
>
> We investigated the training data for XCOMET and confirmed that it does contain Chinese and Japanese data (English-centric), although not specifically from Japanese-Chinese MQM evaluation. Therefore, we believe XCOMET's weaker performance on this pair is not due to a complete lack of knowledge but may stem from:
>
> - a) The Ja-Zh task representing a possible **Out-Of-Distribution** scenario for XCOMET model, testing its generalization limits.
> - b) A potential **imbalance** in the multilingual capabilities of its **XLM-R backbone**.
>
> Notably, our model was also **not trained on any Ja-Zh MQM data**. Its performance likely comes from the superior multilingual and zero-shot generalization capabilities of its underlying backbone model.
>
> ------
>
> **References**
>
> [1] Guerreiro, N. M., Rei, R., Stigt, D. V., Coheur, L., Colombo, P., & Martins, A. F. (2024). XCOMET: Transparent machine translation evaluation through fine-grained error detection. *Transactions of the Association for Computational Linguistics*, *12*, 979-995.
>
> [2] Yan, J., Yan, P., Chen, Y., Li, J., Zhu, X., & Zhang, Y. (2024). Benchmarking gpt-4 against human translators: A comprehensive evaluation across languages, domains, and expertise levels. *arXiv preprint arXiv:2411.13775*.
>
> [3] https://ai.google.dev/gemini-api/docs/thinking#summaries
>
> [4] https://huggingface.co/Qwen/QwQ-32B#usage-guidelines
>
> [5] Guo, D., Yang, D., Zhang, H., Song, J., Zhang, R., Xu, R., ... & He, Y. (2025). Deepseek-r1: Incentivizing reasoning capability in llms via reinforcement learning. *arXiv preprint arXiv:2501.12948*.
>
> [6] Abdin, M., Agarwal, S., Awadallah, A., Balachandran, V., Behl, H., Chen, L., ... & Zheng, G. (2025). Phi-4-reasoning technical report. *arXiv preprint arXiv:2504.21318*.

---

### Note · Authors · 2025-08-13

Dear Reviewers and Area Chairs,

We sincerely thank you for your constructive feedback and engagement throughout the review process. The detailed discussions have greatly strengthened our work. Below, we summarize the key concerns and our responses.

**1. Generalizability and Performance (Reviewers edhE, EpwK, JKag, RoLH).**

- We clarified the difference from XCOMET in MQM training data scale and emphasized our focus on diagnosing and improving LRMs.
- For stronger commercial LRMs, we included new results for Gemini 2.5 Pro, noting that its API limitations on reasoning transparency hinder direct analysis, yet our method remains competitive.
- To assess generalizability, we added experiments on out-of-distribution data (En–Ru) and the latest benchmark for a low-resource pair (Hi–Zh). Results show ThinMQM performs well and is competitive with XCOMET-XXL in these challenging settings.

**2. Methodological Ablation and Stability (Reviewers edhE, EpwK, RoLH).**

- We justified our use of decoding temperature based on prior work and empirical practices, and repeated main experiments multiple times, observing minimal variance that confirms stability.
- Additional ablations on prompting templates and the removal of the Error Span Annotation component confirm the robustness of our approach and the critical role of this component.

**3. Clarity and Presentation (Reviewers JKag, EpwK).**

Some misunderstandings were resolved in discussion. For the remaining issues, we will revise formulations for precision, improve structural self-containment, and ensure figures are clear and methodologically sound.

Through these constructive discussions, we have sought to address the primary technical concerns through extensive new experiments and to clarify key methodological and notational misunderstandings. We humbly believe this work represents the first in-depth analysis of reasoning models for MT evaluation and introduces ThinMQM, a cost-efficient calibration method that improves performance.

Sincerely,

Authors

---

### Decision · Program_Chairs · 2025-09-17

**Decision:**

Accept (poster)

**Comment:**

# Summary
This paper investigates the capability of Large Reasoning Models (LRMs) to serve as evaluators for machine translation (MT) quality, a relatively unexplored topic. The authors identify key challenges in using LRMs for MT evaluation, such as inefficient “overthinking,” scoring overestimation, and material sensitivity. They propose a solution named ThinMQM—a thinking-calibrated MQM scoring method—that aligns LRM reasoning trajectories with human-like annotations using synthetic data. The method significantly improves correlation with human MQM ratings while reducing computation by 35x across multiple LRM scales.

# Strengths
* This is the first systematic study of LRM-as-a-judge for MT evaluation, bridging reasoning models and MT metrics—a timely and significant contribution.
* It proposes an effective way to improve the accuracy of the evaluation metric based on LRM, relying on synthetic data that follow a manual evaluation process.
* The paper is clearly written and well-structured, and includes useful figures/tables to illustrate thinking budgets, score distributions, and model comparisons

# Overall
This paper presents a novel method for evaluating MT output with LRMs. The authors successfully resolved all major weaknesses and questions identified during the author-reviewer discussion period. The paper now offers sufficient contributions to be of broad interest to the research community.